# Basic Approaches to the Design of Intrinsic Self-Healing Polymers for Triboelectric Nanogenerators

**DOI:** 10.3390/polym12112594

**Published:** 2020-11-04

**Authors:** Gulzhian I. Dzhardimalieva, Bal C. Yadav, Sarkyt E. Kudaibergenov, Igor E. Uflyand

**Affiliations:** 1Laboratory of Metallopolymers, The Institute of Problems of Chemical Physics RAS, 142432 Chernogolovka, Moscow Region, Russia; dzhardim@icp.ac.ru; 2Moscow Aviation Institute (National Research University), 125993 Moscow, Russia; 3Nanomaterials and Sensors Research Laboratory, Department of Physics, Babasaheb Bhimrao Ambedkar University, Lucknow 226025, India; balchandra_yadav@rediffmail.com; 4Institute of Polymer Materials and Technology, Almaty 050019, Kazakhstan; skudai@mail.ru; 5Laboratory of Engineering Profile, Satbayev University, Almaty 050013, Kazakhstan; 6Department of Chemistry, Southern Federal University, 344006 Rostov-on-Don, Russia

**Keywords:** self-healing polymers, triboelectric nanogenerator, shape memory polymers, energy harvesting

## Abstract

Triboelectric nanogenerators (TENGs) as a revolutionary system for harvesting mechanical energy have demonstrated high vitality and great advantage, which open up great prospects for their application in various areas of the society of the future. The past few years have seen exponential growth in many new classes of self-healing polymers (SHPs) for TENGs. This review presents and evaluates the SHP range for TENGs, and also attempts to assess the impact of modern polymer chemistry on the development of advanced materials for TENGs. Among the most widely used SHPs for TENGs, the analysis of non-covalent (hydrogen bond, metal–ligand bond), covalent (imine bond, disulfide bond, borate bond) and multiple bond-based SHPs in TENGs has been performed. Particular attention is paid to the use of SHPs with shape memory as components of TENGs. Finally, the problems and prospects for the development of SHPs for TENGs are outlined.

## 1. Introduction

The rapid growth in global energy consumption is leading to the overuse of fossil fuels, which greatly exacerbates the world’s energy and environmental problems. Therefore, the development of green and renewable energy plays a defining role for the sustainable development of the entire human society [1]. One of the priority trends in this area is mechanical energy harvesting as a green and environmentally friendly method, which is a promising approach for powering wearable electronics and sensor networks in the Internet of Things (IoT) and artificial intelligence (AI) [2,3,4,5]. Among the existing methods of harvesting mechanical energy, triboelectric nanogenerators (TENGs) attract special attention, the advantages of which are their low cost, high power density and output performance, light weight, uncomplicated production process, wide choice of materials, high stability, environmental friendliness and versatility in design solutions [6,7,8,9,10,11,12,13,14,15].

Among the methods for improving the performance of TENGs, we should note the functionalization of the surface of triboelectric materials [16,17,18], an increase in the contact area [19,20], optimization of the device design [21,22] and production technologies [23,24]. In addition, the charge pump, circuits, mechanics and manufacturing processes are critical to the further development of TENGs. For example, a strategy has been proposed for a stable long-term increase in TENG output enhancement through the use of a stably pre-charged electret [25]. We also note an increase in peak power of more than 1000 times with high flexibility and high light transmittance (~92%) by artificially introducing electric charges onto the amorphous fluoropolymer [26]. A strategy was proposed for the development of a robust TENG using a cold-rolled metal layer as one of the contact layers. The developed TENG shows no significant degradation in power output even after an extremely high number of operation cycles (~1,500,000 cycles) [27]. A type of TENG employing universal biomechanical energy harvesting with a switchable direction is proposed, which includes a corresponding power transmission unit and a rotating TENG [28]. A lightweight TENG (total weight less than 160 g) has been applied to more than five joints and has been shown to be able to easily generate an output current of more than 10 μA as a result of natural body movements. Of interest is the mechanism of generation of a triboelectric signal during gear-based ordinary power transmission [29]. The generated signal can be used to design a real-time and self-triggering gear-based power transmission monitoring system that is simple, cost-effective and user-friendly. We should also note noted the use of a rationally designed lightweight power transmission unit to improve the output electrical characteristics of the TENG, while minimizing the inconvenience caused by carrying it [30].

However, the choice of tribomaterial is the most effective approach for producing TENG with high output power [31,32,33,34,35]. The necessary properties of tribomaterials are flexibility, extensibility, biocompatibility, lightness, durability, washability and breathability. These qualities are fully possessed by polymeric materials that are widely used in TENGs to achieve high output voltage, current and power [36,37,38]. Although the first TENG was proposed in 2012 [6], to date, extensive experimental data have been obtained on the use of polymers as tribomaterials for TENGs.

It is important to emphasize that TENGs are subjected to constant external mechanical stresses, such as tapping, bending and stretching, during energy harvesting [39,40,41,42]. As a result, the robustness and endurance of TENGs is severely compromised by frequent and unavoidable mechanical damage, such as cracks and eventually rupture, which can lead to malfunction. Such damage easily occurs in modern portable/wearable/flexible devices. Therefore, the development of self-healing TENGs to increase life expectancy is highly desirable [43,44,45,46,47,48,49].

In the last decade, the development of self-healing materials has emerged as a particularly promising way to effectively increase the durability and functionality of TENGs. Self-healing polymers (SHP), characterized by a wide variety of potentially important properties and functions, are the basis of most self-healing materials [50,51,52,53,54,55,56,57,58,59]. Thanks to the efforts of specialists in various fields, in recent years, significant progress has been achieved in the production of such polymers and the development of theoretical principles of their functioning [60,61,62,63,64,65].

Modern SHPs are classified into extrinsic and intrinsic SHPs, depending on whether or not healing agents are used. The self-healing of the first polymers is based on the release of monomers or catalysts stored inside capsules or vessels and present within the polymer matrix, which are immediately released after damage. When mixed, they begin to polymerize to help the cut heal. Even though large-volume self-healing can be achieved, the disadvantages are the slow and single-time (for a capsule-based polymer) healing, and complex manufacturing processes (encapsulation of the monomer and catalyst, followed by their dispersion within the polymer). While significant progress has been made in the field of extrinsic SHPs, there are still drawbacks, including the tedious manufacturing process and limited healing cycles. More importantly, if these complex SHPs are to be introduced into TENGs, the effect on the electrochemical performance and mechanical properties of TENGs is still unknown. As such, no extrinsic SHPs have been reported in TENGs yet.

In contrast, intrinsic SHPs contain reversible covalent and non-covalent bonds, the reconfiguration of which allows the restoration of the polymer’s structure after damage [66,67,68,69]. Covalent dynamic chemistry [70,71,72,73] usually includes disulfide bonds [74,75], imine bonds [76], Diels–Alder (DA) reactions [77,78] and borate ester bonds [79], while non-covalent dynamic interactions [80,81,82,83] include hydrogen bonds (H-bonds) [84,85,86], metal–ligand (M–L) coordination [87,88,89,90,91,92], electrostatic interactions [93,94,95], host–guest interactions [96,97], hydrophobic interactions [98], *π*–*π* stacking interactions [99] and crystallization [100].

The mechanism of intrinsic healing includes the breaking of weak dynamic bonds during damage or under the influence of an external stimulus, and the formation of low molecular weight spices. These spices have a high healing capacity, which contributes to a faster rate of self-healing. At the final stage, the initial structure of the polymer and its functional properties are restored. The process of self-healing can take place many times.

Unlike extrinsic SHPs, intrinsic SHPs are characterized by the functionalization of the polymer by various self-healing groups and multi-time reversible healing. Another advantage of using the intrinsic polymer is the fast healing due to the absence of diffusion and polymerization control steps, which is a decisive factor in their use in wearable devices, where it is necessary to avoid signal interruption due to damage as much as possible. Moreover, due to the need for increased potential uses, intrinsic SHPs have been modified in various ways to achieve high flexibility, fast self-healing, biocompatibility, and many physical (e.g., electrical, electronic and thermal) and chemical (e.g., electrochemical and photochemical) properties. Obviously, intrinsic SHPs are more suitable for TENGs. Because of these advantages, this review focuses on the importance of intrinsic SHPs with specific functions suitable for TENGs. Representative examples highlight recent advances in this area with an emphasis on the relationship between the structure of SHPs and the TENG performance. The main sections of this review include covalent and non-covalent bond-based SHPs. Work on solving important problems in the field of SHPs for TENGs will require the cooperation of scientists from different areas (chemists, materials scientists, bioengineers and physicists), and all of them are aimed at the development of new advanced technologies that contribute to solving global problems of green energy.

## 2. Non-Covalent Bond Based SHPs

### 2.1. H-Bond Based SHPs

Possessing reversibility, directionality and sensitivity, the H-bond is a widely studied supramolecular interaction that plays an important role in many biological processes in nature, such as DNA replication, molecular recognition, protein folding, etc. Due to functional groups (carboxyl, amide, urea pyrimidine, hydroxyl, etc.), which can form H-bonds, H-bond based SHPs are widely used in TENGs. Among these SHPs, various self-healing hydrogels, one of the types of environmentally friendly materials containing a crosslinking network of polymer chains, are of considerable interest [101,102,103]. As a typical example of a self-healing hydrogel for TENG, we note the flexible polydopamine–polyacrylamide hydrogel (PDA-PAM hydrogel) with a self-healing ability (Figure 1) [104]. Specifically, it can self-heal about 10 min after a cut. A simple single-electrode mode TENG with a sandwich configuration was developed, in which the resulting PDA-PAM hydrogel served as the resulting electrode. The hydrogel can significantly enhance the output performance of TENG over traditional copper foil as an electrode, due to advantages such as its superior self-healing ability under ambient conditions, super-stretchability reaching about 6000% before tensile failure, good transparency and high conductivity. The use of a hydrogel as an electrode in flexible TENGs allows relatively stable and excellent electrical output characteristics to be maintained even after strong stretching. Probably, the mechanism of the process is due to the formation of an electric double layer at the interface between the PDA-PAM hydrogel and a copper wire, which promotes the simultaneous transfer of ions in the hydrogel and induces charges.

Among other self-healing hydrogels, we also note polyvinyl alcohol (PVA)/sucrose hydrogel [105], aliphatic polycarbonate hydrogels based on 2-methyl-2-benzyloxy carbonyl propylene carbonate and methoxy poly(ethylene glycol) (mPEG_113_, M_n_ = 5000) [106], three-dimensional network structural gel electrolyte PAA-g-EG_50%_, where the PAA is polyacrylic acid and EG is ethylene glycol [107], and PAA hydrogels with hydroxyl-functionalized basalt fibers as reinforcement [108].

It is of interest to prepare nanocomposite hydrogels. For example, a series of nanocomposite gels was synthesized in a mixed solvent of glycerol–water with inorganic clay as a physical cross-linking agent [109]. The resulting nanocomposite gels exhibited the ability to self-heal through H-bonds (healing efficiency reached 84.8%). In another interesting example, a nanocomposite hydrogel with high real-time self-healing ability was made from silk fibroin, tannic acid and a biocompatible conductive polymer, i.e., poly(3,4-ethylene dioxythiophene):poly(styrene sulfonate) (PEDOT:PSS) [110]. The properties of the hydrogel are mainly due to H-bonds between silk fibroin and tannic acid chains in water. It is interesting to note that the water-induced healing ability of such polymers is reduced by cross-linkers or subsequent acid treatment (Figure 2) [111]. Organic dopants of PEDOT have high water swelling ratios and lead to water-enabled healing, while inorganic dopants fail in the healing of PEDOT. The autonomic self-healing of films prepared from mixtures of conducting polymer PEDOT:PSS and PEG is also observed [112]. The presence of PEG in PEDOT:PSS films decreases the modulus of elasticity and increases the elongation at break, resulting in a softer material with enhanced self-healing characteristics. The healing mechanism is probably associated with the return of the material to the damaged area immediately after the cutting.

It is of interest to use a healable composite based on a thermoplastic elastomer with liquid metal (LM) and silver flakes as a stretchable conductor and a triboelectric layer for TENGs (Figure 3) [113]. The main characteristics of TENG include their 2500% stretchability, their 96.0% recovery of conductivity after healing, and the recovery of their power performance after severe mechanical damage due to the supramolecular H-bond of the thermoplastic elastomer. Another important feature of this TENG is its full stretchable and 3D printing capability, whereby both the electrode and the triboelectric material can be 3D printed.

SHP has also been obtained by the direct patterning of non-toxic and highly conductive LM on PVA hydrogels [114]. Mechanical and electrical self-healing is achieved simultaneously through the use of H-bonds in the PVA hydrogel network and LM fusion (Figure 4A). Experiments on mechanical and electrical self-healing included complete rupture of a PVA hydrogel with a patterned wavy LM line in the LED circuit (Figure 4B). Contact between the two pieces of hydrogel led to the restoration of conductivity within 2 h, after which the repaired circuit retained its functional characteristics even under 60% strain.

Noteworthy is the self-healable nanostructured Ti_3_C_2_MXenes/rubber-based supramolecular elastomer (Figure 5) [115]. Serine-modified MXene nanoflakes with an elastomer matrix create thin dynamic supramolecular H-bond interfaces. The elastomer has the desired recovered toughness (12.34 MJ/m^3^) and an excellent self-healing performance (~100%) at room temperature.

A poly(*n*-butyl acrylate) film functionalized with lanthanum hexaboride (LaB_6_)/poly(methacrylate-2-ureido-4[1H]-pyrimidinone) was prepared to achieve self-healing performance [116]. Through the interaction of H-bonds, the film showed marked progress in the ability to self-healing at ambient temperature. The self-healing rate is about 84% for 20 mg LaB_6_.

A highly efficient flexible TENG with reproducible electronic self-healing characteristics was obtained, in which highly conductive Ag nanowires (NWs) are semi-embedded in a polyurethane/polydimethylsiloxane (PU/PDMS) polymer for the production of all-in-one friction layers, and to improve the self-healing process [117]. The output voltage and current of the healing device can reach approximately 99% of their initial values even after five cutting/healing cycles. The important characteristics of the resulting TENG are excellent stability and flexibility. The TENG self-healing process proceeds under the influence of an external stimulus: infrared (NIR) radiation [118,119]. It is important to emphasize that TPU healing is driven by hot melting, while the PDMS healing process is due to H-bonding. In addition, the manufacture of the electrode involves embedding the AgNWs in the polymer, which prevents the AgNWs from falling out after touching or bending. The result is that the electrode retains good conductivity and stability even after repeated recovery cycles. The resulting TENG can be used to power electronic devices with increased durability and reliability.

Flexible bilayer composite films consisting of healable elastomeric substrates based on n-isopropylacrylamide and 2-methoxyethyl acrylate and wrinkled graphenes exhibit rapid self-healing at room temperature, reaching 96%, due to the reversible H-bonds [120]. The synergy between the self-healing of the substrates and the wrinkled structure of graphene is endowed upon the composite films for mechanical and electrical healing. By adjusting the prestrain ratio of the substrates, the composite films can exhibit customizable self-healing. A cost-effective self-healing composite material was obtained based on a supramolecular organic polymer from dimer acid and diethylenetriamine with microparticles of copper [121]. It has excellent self-healing properties under ambient conditions within 5 min. The healing efficiency is about 90% for its mechanical properties and almost 100% for its electrical properties. A robust self-healing elastomeric matrix of *n*-isopropylacrylamide and 2-methoxyethylacrylate has a polypyrrole (PPy) network of high healing efficiency at body temperature (94%), which is provided by a combination of dynamic H-bond and microphase separation structures during freeze–thaw (Figure 6) [122].

Of interest is the SHP based on PU and PU@multiwalled carbon nanotubes (CNT) on the flexible cellulose nanocrystals@carboxylated nitrile rubber@polyethylenimine substrate (Figure 7) [123]. Due to the self-healing properties of PU and the substrate, as-fabricated SHP has a self-healing ability after damage and maintains excellent sensitivity to temperature and pressure after healing. The healing process mainly occurs through the creation of a large number of reversible dynamic H-bonds.

Multifunctional hydrogels consisting of PVA and PPy nanotubes, prepared using borax as a crosslinking agent, show self-healing properties in 15 s [124]. More interestingly, the hydrogels also exhibited underwater self-healing properties. The dynamic reversible chemical bonds in the hydrogel network are still maintained due to the diffusion of borate in water, allowing hydrogels to heal themselves underwater. However, the self-healing process under water was much slower than that in air, due to the lower concentration of borate and the insufficient contact of hydrogels. Of note is the PDMS elastomer with a self-healing ability (Figure 8a) [125]. Reversible strong and weak H-bonds are included in the system to realize the self-healing ability. After heat treatment at 60 °C for 24 h, the strength of the self-healing efficiency can reach 92.1%. To increase the strength of the PDMS elastomers, a strategy was adopted combining the advantages of hierarchical H-bond dynamics and a phase-separation-like structure [126]. By modifying both the stronger and weaker H-bonds within the PDMS network, a number of autonomous SHPs have been achieved (Figure 8b). The transparent TENG was made by sandwiching a hydrophilic polymeric material as a conductive layer between two pieces of the PDMS elastomer. Thus, a self-healing (~97% efficiency) TENG was built.

Note also the class of SHPs cross-linked through rationally designed interactions of H-bonds with different strengths (Figure 9) [127]. The resulting supramolecular network in the polymer film realizes autonomous self-healing.

An SHP was synthesized, consisting of crystalline poly(l-lactic acid) domains, a reversible H-bond region, and a poly(tetrahydrofuran) soft chain (Figure 10a) [128]. The mechanical properties of the crystallization-containing SHPs can be fully restored after self-healing. Nanofibrous hard domains induced by hierarchical H-bonds and crystallization structures of the polymer coexist in a soft matrix. We are interested in SHP based on 2,4′-tolylene diisocyanate, isophorone diisocyanate and poly(oxy-1,4-butanediyl), characterized by the targeted regulation of the molecular segment’s hardness and dynamic crosslinking strength (Figure 10b) [129]. The dual H-bond from the 2,4′-tolylene diisocyanate unit is a relatively strong cross-linking bond, and it determines high mechanical strength. The weaker single H-bond from the isophorone diisocyanate unit promotes the efficient dissipation of strain energy as a result of bond cleavage, which serves as the basis for the rapid self-healing of the polymer at room temperature. The result is polymers with high self-healing efficiency (97% within 6 h).

The self-healing PU was prepared by constructing a dynamic cross-linked network through quadruple H-bonds formed from ureidopyrimidinone (UPy) [130]. SHP showed excellent self-healing properties and was able to fully recover its original mechanical strength after being repaired at 80 °C for 24 h. UPy was grafted into the side chain of waterborne PU to impart a self-healing function to the polymer (Figure 11) [131]. Methylated β-cyclodextrin, which was capable of enhancing the mechanical properties of PU and realizing a synergistic effect between cyclodextrin and UPy to enhance the self-healing efficiency, was introduced into the main chain of PU. The UPy side chain formed quadruple H-bonds synergistically with the host–guest interaction between UPy and cyclodextrin moieties to achieve the self-healing properties of the PU. When destruction occurred on the PU, the polymer segment began to move simultaneously with the dissociation/recombination of the H-bond and the dynamic balance of host–guest recognition under heating conditions. The cut polymer spline showed a self-healing efficiency of 92.29% at 100 °C for 36 h.

The double-network (DN) structure system has been shown to be effective in producing a healing PU elastomer [132]. The weakly cross-linked chemical network in the DN system based on a photocurable double acrylic bond creates a strong molecular framework and determines the elasticity of the PU. The physically cross-linked network based on the quadruple H-bonds of the UPy elements provides fast reformation after fracture and dissipates strain energy. This weak dynamic bond determines the excellent self-healing ability (90%) and high stretchability of the elastomer. By introducing commercially available guanine and cytosine into PDMS, SHP can be achieved [133]. A base pair functionalized PDMS-based elastomer has been successfully developed, and exhibits self-healing properties that can reach 75% efficiency. The SHP species, CO_2_-based polyureas, can be recovered by up to 95% at room temperature [134]. These unexpected properties are the result of the rich and diverse H-bonds in the polyurea molecular chain between urea fragments. The strength, arrangement and density of H-bonds are controlled by adjusting the amount of the starting monomers, and thus self-healing properties can be optimized. Self-healing PDMS-based elastomers were obtained as a result of the reversible destruction and re-formation of supramolecular interactions (Figure 12) [135]. A library of poly(urea)s for studying self-healing was obtained by changing the ratio between the starting monomers and changing the molecular weight of the PDMS precursor. Thus, an optimum balance was achieved between mechanical properties and self-healing performance, and due to an additional reduction in the molecular weight of the precursor polymer, a minimum recovery of 80% in stress was observed within 12 h at room temperature.

The unique macromolecular structure and sufficient amount of H-bonds of the lignin-*g*-poly(*n*-butyl acrylate-*co*-1-vinylimidazole) copolymers resulted in exciting and controllable characteristics (Figure 13a) [136]. When the 1-vinylimidazole content was increased to 19.4 wt. %, the elastomer automatically showed self-healing at room temperature. By incorporating a sliding cross linker (polyrotaxanes) and H-bonds into the polymer, an SHP was obtained that can achieve almost complete self-healing (93%) at 55 °C (Figure 13b) [137].

A supramolecular polymer material is made by topologically limiting the biodegradable linear polymer of PVA through naturally occurring dendritic tannic acid molecules based on high-density H-bonds [138]. The biodegradable composites are healable after break at room temperature with the assistance of water to activate H-bond reversibility. A polyamide elastomer was prepared from dimer acid and diethylene triamine with a fast self-healing ability [139]. Numerous H-bonds formed between the amide bonds of different segments allowed elastomer’s self-healing at room temperature. The elastomer’s self-healing efficiency can reach 80% within 2 h. SHP, which can simultaneously have high flexibility, transparency and a special self-healing ability, is obtained through the rational combination of PSS with PVA [140]. Even being cut off, SHP can heal spontaneously and restore its flexibility, transparency and electric power with 100% healing efficiency. The self-healing ability of SHP is explained by the restoration of the damaged PSS-PVA film due to H-bond interactions, as well as the reconnection of the Ag NW layer due to the van der Waals forces. Under the influence of moisture, the SHP crack (Figure 14, left) will gradually fuse together due to H-bond interactions between the PSS-PVA chains and water molecules (Figure 14, middle). The PSS-PVA film returns to its original state after drying (Figure 14, right).

SHPs were obtained and rationally grafted with oligo poly(ethylene glycol) [141]. The network, based on the formation of multiple weak H-bonds, endows the elastomers with good complex characteristics, including a fast self-healing process (40 s). We should note the design of self-healing magnetic nanocomposites prepared from readily available commercial monomers: acrylamide and *n*-butyl acrylate [142]. They demonstrate a self-healing ability, reaching a 46% recovery of extensibility in 5 h under ambient conditions. Acrylamide was introduced to present a dynamic self-healing motif with its H-bond amide group. Self-healing PDMS-g-PU/V_2_O_5_ nanofiber supramolecular polymer composites based on the reversible H-bond mechanism were prepared [143]. The healing efficiency of the composite increases with the healing time and reaches 85.4 ± 1.2% after waiting for 120 min at 50 °C. The addition of V_2_O_5_ nanofibers increases the healing efficiency of PDMS-g-PU. In particular, an increase in the healing efficiency from 85.4 ± 1.2% to 95.3 ± 0.4% is observed after the addition of 10 wt. % of V_2_O_5_ nanofiber to the polymer. A self-healing hydrogel is designed with a semi-interpenetrating network formed by the incorporation of carboxymethyl chitosan and sodium chloride into the PAM network (Figure 15a) [144]. Of interest is a macromolecular elastomeric gel with unique capabilities not only for self-healing, but also for transient properties at room temperature (Figure 15b) [145]. By incorporating small molecules of glycerol into hydroxyethylcellulose, a self-healing macromolecular elastomeric gel is obtained. Dynamic H-bonds are formed between the hydroxyethylcellulose chain and the guest small molecules of glycerol, which gives the elastomer the ability to self-heal under ambient conditions.

Of note is the fully self-healing TENG obtained by injecting PDMS-PU-based SHPs and electrodes consisting of small magnets into the device (Figure 16) [45]. Both the open circuit voltage and short-circuit current of a healed device can reach over 95% of their initial values, even after the fifth breakage-healing cycle. The mechanical properties of PDMS-PU and the electrical properties of magnetic-assisted electrodes can be restored by simply connecting the broken ends. The device demonstrates an adaptability to shape in accordance with applied mechanical stimuli for the efficient harvesting mechanical energy. However, it was neither transparent nor stretchable, and heating was required for recovery.

Various PUs are advanced polymers for TENGs. The introduction of different functional groups into the PU molecule leads to an additional improvement in self-healing properties. For example, a functional PU based on a heterocyclic group demonstrates better self-healing efficiency than a conventional PU without heterocyclic groups [146]. These results are attributed to a unique supramolecular network resulting from the strong H-bond interaction between the urethane group and the heterocyclic group in the PU matrix (Figure 17a). Noteworthy is the impregnation with ionic liquids (ILs) of a mechanically robust PU network, consisting of crystallized poly(ε-caprolactone) (PCL) and flexible PEGs, which are dynamically cross-linked by hindered urea bonds and H-bonds (Figure 17b) [147]. This design gives the resulting ionogels excellent healing properties. In particular, SHPs can be easily healed by heating to 65 °C, which restores their original ultra-durable sensitivity. The long-term durability of SHP is attributed to a combination of non- volatile ILs, excellent healing properties and well-designed mechanical properties.

One of the interesting approaches to the creation of SHP is the introduction of polypropylene glycol (PPG) units with high-mobility segments and different lengths at the ends of their PU chains [148]. The increase in the length of the PPG units first leads to an increase in their self-healing ability, and then to a reduction in this ability. The results obtained are related to the combined effect of the mobility of the PPG segments and the density of Н-bonds. In the original PU sample, H-bonds are formed between the carbonyl groups of the urethane, or carbonyl groups in the PCL segments and NH-groups of the urethane (Figure 18, left). In PU samples containing PPG terminal blocks, the formation of H-bonds occurs between the carbonyl groups and the NH-groups of the urethane. At the same time, the formation of H-bonds is impossible between the PPG segments and urethane NH-groups (Figure 18, right). As a result, the density of H-bonds in pure PU is higher than in the PU containing the PPG end blocks.

Of interest as well are diketopyrrolopyrrole-based “alternating” copolymers with urethane-containing side chains, which have self-healing properties with crack healing and the recovery of electrical properties after treatment (Figure 19) [149].

### 2.2. M–L Bond Based SHPs

M–L bond-based SHPs are a widespread type of polymers for TENGs. As a typical example, we note the use of two different ligands of Fe^3+^ ions to obtain a dual-cross-linked polymer network PDMS containing strong and weak M–L bonds (Figure 20) [150]. A weaker bond is formed in the case of pyridinedicarboxamide, and serves to efficiently dissipate energy during cleavage, exchange or transformation. At the same time, stronger bonds are formed in the case of *N*-acetyl-l-cysteine to maintain the integrity and elasticity of the network. The synergistic effect of strong and weak M–L bonds allows the elastomer to self-repair at room temperature. The ratio of these bonds can be controlled by the amounts of added Fe^3+^ ions, which makes it possible to regulate the ability to self-repair.

A rational design of elastomeric vitrimers by embedding Zn^2+^–imidazole complexes into the network has been demonstrated [151]. M–L bonds act as sacrificial blocks for reversible fracture and transformation, resulting in a large increase in parameters such as modulus, strength and toughness, while maintaining the elasticity of the networks. The mechanism of the process includes the breaking of the complexes upon stretching to form covalent bonds, as well as their reversible breaking and re-formation, during which tremendous energy is dissipated and the integrity of the material is preserved. Imidazole-zinc bonds were also incorporated into the cross-linked silicone elastomer to generate SHPs [152]. Reforming the M-L bonds did indeed promote rupture-healing (Figure 21a). Of interest is a system of self-healing interpenetrating polymer network based on chemically/ionically cross-linked PAA containing ferric ions intercalated with physically cross-linked PVA [153]. Ionic bonds are formed between ferric ions and carboxyl groups of PAA, which act as dynamic bonds within the gel networks (Figure 21b). This hydrogel is characterized by a good self-healing ability, i.e., a recovery of original mechanical properties to over 80% after 24 h and 0.14 S m^−1^ ionic conductivity, which is electrically healable.

SHP was obtained by crosslinking a polymer containing azopyridine ligands with Zn(II) ion (Figure 22a) [154]. The resulting photo-healable polymer has a high elongation (up to 735%), high toughness (16.33 MJ/m^3^), and an excellent light-healing efficiency of ~90% after three cutting/healing cycles at moderate temperature (40 °C) under various harsh conditions (e.g., underwater, subzero temperature). Of interest are SHPs, which can be healed by heating (Figure 22b) [155]. They consist of a mixture of linear and three-arm branched PUs with triazole ligand end groups that are non-covalently linked via Fe–triazole interaction. Due to the dynamic nature of the Fe–triazole interaction, all metallopolymers demonstrate excellent healable efficiency (over 90%) depending on toughness.

SHPs are made by uniformly dispersing nanocomposites of cellulose nanofibers and graphene in a PAA hydrogel (Figure 23a) [156]. The resulting SHPs exhibit a healing efficiency of 96.7% within 12 h. Their self-healing ability is due to the M–L complex interactions between the Fe^3+^ ions and carboxylic groups of PAA and cellulose nanofibers. A hierarchical strategy for temporarily controlling the intrinsic healability of kinetically inert and highly stable metallopolymers is presented (Figure 23b) [157]. Enzyme-regulated competing reactions are used to temporarily program changes in the oxidation state of metal ions in the cobalt cross-linked polymer hydrogels, which, in turn, tune the M–L interactions for the effective healing of defects and the restoration of hydrogel properties. The hydrogels show excellent healability with ~100% recovery.

A physically cross-linked hydrogel with poly(acrylamide-co-acrylic acid)-Fe^3+^ and chitosan-SO_4_^2-^ dual ionic networks has been developed [158]. The optimal hydrogel has the property of fast self-healing. The self-healing of stretchable hydrogels should also be noted; these consist of a pluronic F127 diacrylate cross-linked PAA network and PDA, and are additionally cross-linked with Fe^3+^ [159]. The unique structure provides the resulting hydrogels with excellent self-healing properties. A starch/PAA hydrogel was synthesized by simply loading ferric ions (Fe^3+^) and tannic acid-coated chitin nanofibers into the hydrogel network (Figure 24) [160]. The 1 wt. % reinforced hydrogel showed a healing efficiency of 96.5% after 40 min.

## 3. Covalent Bond Based SHPs

The inclusion of weak non-covalent bonds in polymers leads to the fact that the SHPs based on them are mechanically unstable in comparison with polymers based on covalent bonds. Therefore, creep is a big problem for such SHPs, since weak non-covalent bonds often undergo various transformations even under normal mechanical stress. Compared to reversible non-covalent bonds, reversible covalent bonds have a higher bond energy, which is useful for creating strong bonding between molecules. Therefore, these bonds are widely used to create SHPs.

### 3.1. Imine Bond Based SHPs

It is of interest to use a dynamic imine-bond based elastomer to create a transparent triboelectric layer that can be healed at room temperature [44]. Such bonds are introduced into the PDMS network to repair mechanical damage (94% efficiency), while the mechanical elastomer recovery is transferred to the curved electrode for electrical healing. TENG can restore power generation (100% healing efficiency) even after accidental cutting. Unfortunately, the performance of this TENG was limited by its rigid and opaque electrode (Ag NWs and PEDOT composite); as a result, the TENG achieved only 50% stretchability and ≈73% transmittance, and took ≈12 h for healing. Among other interesting examples, we note a simple approach to making PUs with self-healing properties that are imparted by dynamic urea bonds in the matrix (Figure 25a) [161]. A transparent healable PDMS elastomer based on imine bonds has been successfully prepared from an amino-modified PDMS and 1,4-diformylbenzene (Figure 25b) [162]. The reversible imine bonds based on the Schiff base linkage are a critical structural factor that affects the self-healing ability, which can promote fracture surface contact and result in rapid and repeatable self-healing without any external stimuli, such as heat or light at room temperature.

Typically, the T_g_ of polymers is an important factor for autonomous self-healing; in particular, a lower T_g_ promotes the better mobility of polymer chains, leading to chain diffusion, bond exchange, and re-entanglement on broken surfaces [163]. In addition, an increase in temperature also increases the efficiency of healing, as with most self-healing materials [164]. The inclusion of aromatic moiety-containing diamine-based hindered urea bonds can significantly improve the thermal and mechanical characteristics of PUs, while maintaining the desirable self-healing (Figure 26) [165]. It is important to emphasize that PUs regained much of the mechanical strength and integrity of the original material after restoration. At the same time, a three-dimensional cross-linked PU without a hindered fragment of a urea bond does not possess a self-healing ability due to the high stability and irreversible nature of the chemical bonds.

The SHP obtained as a result of the condensation reaction of the Schiff base demonstrates effective self-healing with increasing temperature; in particular, the recovery of mechanical properties was 96% after 2 h of healing at 80 °C [166]. This result is due to two reasons. First, the symmetric structure of the imine-diol in combination with the rigid C=N double bond contributes significantly to the microphase separation and orientation of chain segments during crystallization under stress. Secondly, in the hard phases, the dynamic reaction of the dynamic exchange of imine is phase-locked. It should also be noted that nanocomposites of thermoplastic PUs and PPy were obtained by solution blending [167]. After the composite film was cut and spliced, the fracture was irradiated with NIR, and the mechanical strength of the composite material could be restored to more than 80% in only 30 s.

SHP based on acylhydrazone bonds was prepared from environmentally friendly polysaccharide materials [168]. It can achieve rapid self-healing within minutes after cutting, and has properties such as being free-shapeable. An ionic gel was obtained on its basis by introducing Fe_3_O_4_ to the gel precursor, which can instantly realize self-healing due to the combination of the acylhydrazone bond with the M–L bond (Figure 27a). Moreover, the strength of ion gel is more than 10 times greater than before. Of note is the introduction of dynamic covalent chains, which are composed of three or five dynamic polyurea bonds, into a cross-linked PU [169]. Compared to the dynamic urea bond, the inclusion of multiple dynamic polyurea chains results in a significant increase in the healing rate/efficiency of the network. These results can be understood from their different reaction mechanisms (Figure 27b). Normal healing requires the diffusion and collision of reactive groups at the end of the chain, which is largely determined by the mobility of the polymer chain. On the other hand, after the dynamic covalent chain’s collapse, the generated small fragments have high mobility and reactivity, enabling them to easily diffuse into the damaged region and perform healing, forming a dynamic covalent chain again.

Of interest are chemically cross-linked ionogels with excellent self-healing properties based on gelatin synthesized via the Schiff base reaction in IL of 1-ethyl-3-methylimidazolium acetate [170].

### 3.2. Disulfide Bond Based SHPs

Among the commonly used reversible covalent bonds for SHP production, disulfide bonds have received more attention in recent years due to their outstanding advantages. First, disulfide bonds have relatively high bond energy, which can promote the formation of strong bonds between molecules and improve the healing efficiency of polymers. Second, a reversible disulfide bond exchange reaction can be initiated at a relatively low temperature, and materials can be healed under relatively mild conditions. For example, a remote, fast and efficient self-healing PU nanocomposite was obtained, in which CNTs were embedded in disulfide bond-based PU [171]. The dynamic disulfide exchange reaction could be stimulated by NIR irradiation, and the damage to the composite could be healed in minutes due to the photothermal effect of CNTs. The healing efficiency was more than 80% after 1 min of NIR irradiation. Self-healing PU films have been developed by creating reversible cross-linked networks with a particular synergistic effect between lithium salt and IL [172]. As the load of lithium salt and ionic salt increases, the films show increased self-healing properties. The design of reversible covalent cross-linking provides PU films with a high self-healing efficiency after complete fracture (92.7%) and a high recovery of mechanical properties after remolding (91.8%). Of note is the self-healing of TENG induced by NIR radiation [43]. The developed TENG contains three layers: PDMS (triboelectric layer), an epoxy-based polysulfide elastomer (a layer that promotes healing), and CNT (a layer of self-healing electrostatic induction electrode). The heat rapidly generated by a remotely induced dynamic disulfide exchange reaction, stimulated by NIR radiation, is transmitted from the bottom up, and initiates a zipper-like self-healing process. It is important that the electrical characteristics can be restored even after five repairs. In addition to the high self-healing temperature (150–160 °C), the sample can also be healed at a medium temperature (~80 °C) in a few minutes, suggesting that the self-healing temperature is regulated in a manner dependent on the practical application (in vivo or in vitro). An important feature of the developed devices is the ability to scale, like LEGO blocks, by touching the side faces under the influence of NIR irradiation, or an increase in electrical power. Interestingly, the self-healing of TENG has been shown in chicken skin due to the high penetration of NIR into biological tissues, opening up the prospects for their use in creating potential implantable electronics.

A fast self-healing siloxane elastomer synthesized by incorporating aromatic disulfides into a siloxane matrix showed a healing efficiency of more than 95% at room temperature (Figure 28) [173]. The healed siloxane elastomer was able to recover 357 ± 15% of its elongation at break after healing within just one minute at room temperature.

A TENG based on a vitrimer with a dynamic disulfide bond with a tensile strain of ≈100% was obtained by embedding a layer of Ag NW percolation network into an elastomer [39]. Thanks to the covalent dynamic disulfide bonds in the elastomer matrix, a thermal stimulus provides in situ healing in the case of breakage, a change in shape configuration along with demand, and the assembly of more complex designs of TENG devices. Due to its excellent ability to restore dynamic disulfide bonds between fracture surfaces, an almost 100% efficiency of vitrimer healing can be obtained here. This self-healing and shape-adapting TENG has been shown to be useful in mechanical energy harvesters and self-powered tactile/pressure sensors, with extended service life and superior design flexibility. However, it was not transparent, and required heating to 65 °C for several hours to recover from damage. A self-healing and reproducible disulfide-based liquid crystal elastomeric composite was prepared by incorporating graphene nanoplate fillers. The composites exhibited intriguing recycling characteristics (tensile strength after recycling could be retained at over 93% compared to the original composites) [174]. Evidently, superior self-healing was associated with the exchange chemistry of disulfide dynamic covalent bonds under heating conditions (Figure 29).

A self-healing thermoplastic PU based on an aliphatic disulfide bond with a distinct two-phase morphology was manufactured, showing high healing efficiency at moderate temperatures [175]. It consists of stable and inert soft segments and active hard segments with dynamic aliphatic disulfides and strong H-bonds. A polyacrylate coating with a strong and flexible crosslinking network has been developed that exhibits a high self-healing efficiency in a short time [176]. SHP has a high self-healing efficiency of 87.6% after 2.5 min of NIR laser irradiation. The high mechanical strength and high self-healing efficiency are due to the reversible covalent cross-linking between hyperbranched polysiloxane with multi-maleimide end groups. As the temperature rises, some of the DA reversible covalent cross-links break, which leads to an increase in the mobility of the polymer chains and facilitates the self-healing process (Figure 30).

The silicone self-healing elastomer was obtained by sequential thiol-ene UV-curing between thiol and vinyl functionalized polysiloxanes, and thermocuring between carboxyl and amido functionalized polysiloxanes [177]. These resulting SHPs exhibit an excellent 97% healing efficiency. The resulting elastomers also exhibit an excellent 97% healing efficiency. Moreover, these elastomers can be reprocessed to recover 90% of their original mechanical strength, while recycled elastomers can still repair damage with over 90% efficiency.

### 3.3. Borate Bond-Based SHPs

Reversible B–O bonds have great potential for use in SHP creation [178]. For example, linear SHPs based on trigonal planar boronic esters undergo reversible depolymerization due to the hydrolytic cleavage of boronic ester bonds in their main chain [179]. The use of dynamic crosslinks based on boronic ester makes it possible to obtain three-dimensional bulk polymer networks, which are self-healing at room temperature (3–4 days) [180]. This long-term healing process involves the usual chain diffusion through the cut interface and subsequent bond reformation [181]. A nanocomposite organohydrogel was prepared from the conformal coating of a functionalized reduced graphene oxide network consisting of PVA, phenylboronic acid grafted alginate, and PAM in a binary EG solvent system (Figure 31) [182]. The nanocomposite organohydrogel manifests self-healing properties without any other external stimuli.

PVA-based hydrogels with outstanding biocompatibility and excellent mechanical properties can be used as a matrix for wearable TENGs [183]. The inclusion of photothermally active PDA particles and multiwalled carbon nanotubes (MWCNTs) allows the TENG to physically self-heal itself after 1 min when exposed to NIR radiation. At the same time, the chemical self-healing of TENG can be triggered by spraying water at a temperature of 25 °C, with the introduction of water-active dynamic borate bonds into the hydrogel (Figure 32). In addition, the applicability of TENG as a soft energy device for harvesting the energy of human movement has been demonstrated. In particular, tapping TENG with various deformations, the rectified electricity can charge industrial LEDs with sustainable energy. When operating in single electrode mode, the electrical outputs of TENG remain stable even at 200% deformation, since the MWCNTs are uniformly dispersed in the matrix and act as conductive fillers in TENGs.

Of note are agarose/PVA double network hydrogels, which have the advantages of self-healing and tolerability to several types of damage [184]. By integrating the double network hydrogel substrate, the SHP’s original efficacy can be restored to 90% after five healing cycles, and each rapid healing time is shortened to 10 s. The self-healing mechanism is based on dynamic covalent bonds between the complexation of PVA hydroxyls and borate ions.

## 4. Multiple Bond-Based SHPs

The healing efficiency of waterborne PUs containing dynamic aromatic bonds of Schiff bases is 83.8% (commercial LED table lamp, 24 h~25 °C) [185]. It is important to emphasize that these SMPs exhibit a good balance of self-healing at room temperature and high mechanical properties. This self-healing is due to the imine metathesis of aromatic Schiff base bonds and H-bonds between urethane groups, the first factor being the main one (Figure 33).

Of interest is transparent PU, which self-heals at room temperature with the aid of water (Figure 34) [186]. It should be noted that the mechanical properties of PUs are determined by both imine bonds and the density and strength of hierarchical H-bonds within the PU networks. More importantly, these two types of bonds promote repeatable self-healing at room temperature with a high healing efficiency (92.2%).

An approach has been proposed for the synthesis of room-temperature self-healing PDMS elastomers [187]. Due to dynamic intermolecular H-bonds, reversible imine bonds and highly flexible Si-O chains, the elastomer showed excellent self-healing properties with a healing efficiency of 95% at room temperature for 24 h. Even with the use of water and artificial sweat, the healing efficiency reached 89% and 78%, respectively. Silicone elastomers with adjustable self-healing properties have been prepared [188]. They contain characteristic imine and urea groups that determine the self-healing ability. The dynamic interaction of imine groups decreases with an increase in the number of urea groups due to the limited movement of the polymer chain by H-bonds, which leads to a deterioration of the self-healing properties (Figure 35). First, the elasticity of the polymer forces the broken ends to come closer to each other, and then the rapid recombination of H-bonds between urea groups binds the broken ends more closely. Finally, a Schiff base reaction occurs between the aldehyde and amino groups in the damaged parts, which leads to the complete self-healing of the polymer.

Fragments containing reversible imine bonds and quadruple H-bond (UPy) are injected into polymer networks to create a self-healing electrification layer (Figure 36) [189]. UPy-functionalized MWCNTs are then incorporated into SHPs to form a conductive nanocomposite. The resulting materials, created through dynamic bonds, exhibit excellent intrinsic self-healing and shape memory properties. Moreover, TENGs achieve robust interface bonding thanks to the similar recoverable networks between the electrification layer and the electrode. The output electric performances of the self-healing TENG can almost reattain their original state when the devices are damaged. In addition, the self-healing TENG has been applied in self-powered devices to detect human movement. It is important that thermal IR radiation from human skin leads to effective self-healing after damage. This allows the integration of self-healing and self-powered software devices, resulting in reliability, safety and environmental sustainability.

A self-healing PU elastomer was obtained in which the restoration of stretchability and mechanical sensitivity was achieved by introducing multiple disulfide and H-bonds into the molecular chains of PU (Figure 37) [190]. The self-healing mechanism at moderate temperatures involves a reversible dynamic bond reaction and reconstruction over the entire damaged surface. In addition, the polymer network will be recross-linked due to the chain migration in PU, and the process can be influenced by temperature. It is important that the self-healing of the PU elastomer can be repeated many times due to the dynamic nature of the dynamic bonds and polymer chains.

Of interest is a self-antiglare waterborne PU that has excellent robust mechanical properties when working with self-healing materials at room temperature (Figure 38a) [191]. For this purpose, an aromatic disulfide, containing two amino groups that can provide a unique quadruple H-bond and a disulfide bond, was included in the hard segment. Compared to the linear-array H-bond, the unique H-bond of the zigzag array can be effectively exchanged at room temperature, causing nearby disulfide metathesis. Thus, the mechanical properties can be restored to more than 83% at room temperature after the sample has been cut in half and reconnected. Note also the use of PVA-g-PCL-cured, isocyanate-terminated, disulfide-containing PU to tune the self-healing efficiency (Figure 38b) [192]. The dynamic disulfide bonds and Н-bonds in this system contribute significantly to the self-healing behavior at moderate temperatures. The self-healing efficiency is, impressively, as high as 94%.

Studies of such SHPs based on disulfide and H-bonds are numerous [193,194]. We ought to note the PU containing a polydisperse hard segment with H-bonds, a hydrophobic soft segment, and a dynamic disulfide bond (Figure 39a) [195]. PU is characterized by a high density of self-healing points along the main chain and a faster self-healing speed, which reached 1.11 μm/min in a cut-through sample and recovered more than 93% of its original mechanical properties in 6 h at room temperature. The self-healing thermoplastic PU includes both a soft segment (polytetramethylene ether glycol) and a hard segment based on aliphatic isocyanate (m-xylylene diisocyanate) and aliphatic disulfide (bis(2-hydroxyethyl)disulfide) (Figure 39b) [196]. The combination of dynamic disulfide and H-bonds allows the PU elastomer to self-repair with a self-healing efficiency of about 39% at moderate temperatures.

Of interest is a solid polymer electrolyte containing disulfide bonds and urea groups (Figure 40) [197]. The H-bond between urea groups and the disulfide metathesis reaction provides SHP with a high level of self-healing without external stimuli at room temperature, as well as ultra-fast self-healing at elevated temperatures. Fully healed SHPs with extreme damage show high self-healing efficiencies.

SHP should be considered, given that it demonstrates a high (>80%) self-healing efficiency (after ≈24 h) in high humidity and/or different (under)water conditions, without the help of external physical and/or chemical triggers [198]. Soft electronic devices based on this SHP demonstrate high reliability and the ability to recover their electrical properties after damage both in the environment and in water. Universal SHP is designed by synergistically incorporating multi-strength H-bonds and disulfide metathesis into PDMS polymers (Figure 41) [199]. The self-healing process proceeds differently under various conditions, such as 10 min at room temperature and at ultra-low temperatures (−40 °C), under water with a healing efficiency of 93%, in supercooled highly concentrated salt water containing a 30% NaCl solution at −10 °C with a healing efficiency of 89%, and in a strong acid/alkali environment at pH = 0 or 14 with a healing efficiency of 88 or 84%, respectively.

Some interesting representatives of SHPs include recycled PU with a high self-healing ability (100%, 6 h, 25 °C) and a smart PU composite based on CNTs decorated with PPy [200]. Various damages (broken, scratches, cracks, etc.) contribute to the breaking of bonds in the main chain of PU and points of mutual contact of PPy fragments at the interfaces of the conductive materials PPy/CNT. Broken bonds can be repaired by disulfide and H-bonding due to the superior mobility of the PU matrix chains, resulting in the reconnection of the PPy interconnection points.

We should note SHPs based on dynamic diselenide and dual H-bonds, which simultaneously contain flexible fluorinated siloxane units and H-bonded urethane and urea groups (Figure 42) [201]. The self-healing process of these SHPs occurs at room temperature for 2 h in visible light. The healing of the prophase after the reconnection of the fracture surfaces is due to the restoration of the dual H-bonds of the urethane and urea moieties, and the diselenide metathesis, triggered by visible light, is facilitated by macromolecular chains containing fluorinated siloxane units with excellent flexibility.

Similar SHPs with superior healing properties were developed by combining visible light-induced dynamic diselenide bonds and pendant UPy groups [202]. The synergistic effect of these dynamic bonds was critical for the rapid healing process (10 min) of polymers in visible light with a healing efficiency of more than 95.0%. Thanks to this sophisticated architecture, rapid self-healing and re-processing at room temperature can be achieved in super-tough polymers (69.10 MJ m^−3^) with eco-friendly characteristics.

Of interest are the visible light-induced SHPs with outstanding healing properties based on dynamic ditelluride bonds and UPy fragments in backbones (Figure 43) [203]. These SHPs showed rapid recovery due to the physical cross-links formed by quadruple H-bonded UPy fragments. Rapid self-healing in visible light (85.6% healing efficiency in 10 min) can be achieved by adjusting the ditelluride and UPy content.

A simple method is proposed for the preparation of a chitosan–polyoxometallate hydrogel with outstanding self-healing properties [204]. Of note is a ternary polymer composite composed of polyaniline (PANI), PAA and phytic acid, which can exhibit excellent self-healing properties [205]. In the case of rupture, the electrical and mechanical properties can be restored to an efficiency of ≈99% within 24 h, which is facilitated by dynamic H-bond and electrostatic interactions (Figure 44).

Similar SHPs were obtained, consisting of a conjugated PANI polymer, a non-conjugated anionic polyelectrolyte (poly(2-acrylamido-2-methyl-1-propanesulfonic acid)) and phytic acid, in which the three components dynamically interact via H-bond and electrostatic interactions [206]. The proposed SHPs demonstrate autonomous self-healability without any external stimuli. The SHP performance was stable under severe deformation (50% strain) and multiple self-healing processes (30 cycles). An SHP based on (3-dimethyl (methacryloyloxyethyl)ammonium propane sulfonate–PAA/H_2_SO_4_/bromamine acid sodium, cross-linked by reasonably engineered H-bonds between carbonyl and hydroxyl groups and ionic associations between sulfonic acid and quaternary ammonium groups, was prepared (Figure 45a) [207]. The resulting SHP exhibited significant self-healing in just 8 min. The SHP derived from intermolecular networks between PAA and reduced graphene oxide (rGO) exhibits both electrical and mechanical self-healing properties (Figure 45b) [208]. After cutting, SHP rapidly (~30 s) and effectively (~95%) healed at room temperature.

Of interest are imidazolium-functionalized ionic PUs with a fast self-healing ability [209]. The synergistic effects of H-bonds and reversible ionic interactions between ionic pairs facilitate the auto-repair of cracks at moderate temperatures (40 °C). It is important that an increase in the length of the alkyl chain promotes better self-healing. In addition, ionic PUs have a good shape memory with fast shape fixing and recovery rates. SHP was obtained by combining a physically cross-linked gellan gum network with a chemically cross-linked PAM network [210]. The incorporated H-bond and ionic associations in the gellan gum network, acting as sacrificial bonds, endowed the SHP with excellent and controllable mechanical properties.

SHPs were obtained based on MWCNTs dispersed in thermally reversible cross-linked polyketones [211]. The reversible nature is based on both covalent (DA) and non-covalent (H-bond) interactions. The self-healing effect caused by electricity has been qualitatively demonstrated in the repair of microcracks.

A solvent-involved cross-linking system of PAA, PVA, borax, EG and water has been proposed that is capable of antifreezing below −90 °C [212]. This gel has shown an excellent, fast and effective self-healing ability, which is due to the multiple reversible bonds (Figure 46). Borate bonds of the PVA-B and EG-B type quickly form between adjacent EG-tethered PVA chains when the fracture surfaces touch, resulting in a fast self-healing process. We should also note the restoration of multiple H-bonds between PAA, PVA and EG at the interface, which also facilitates the self-healing process.

The hydrogels based on PANI coated on the surface polycarboxylic multi-branched cellulose nanocrystals served as a dynamic bridge, which endowed these hydrogels with a hierarchical structure and dynamic H-bond interactions doped with the PVA/borax system [213]. In combination with the dynamic borate ester bonds, nanocomposite hydrogels exhibit rapid self-healing. In particular, hydrogels can maintain a good self-healing ability both in air and underwater without any stimuli, and the self-healing efficiency can reach 99.56% within 120 s.

The self-healing MXene nanocomposite organohydrogel is designed by immersing the MXene nanocomposite hydrogel in EG solution to replace part of the water molecules (Figure 47a) [214]. The prepared gel exhibits a superior self-healing ability. A self-healing TENG has been reported, in which a viscoelastic polymer (Figure 47b), commonly known as Silly Putty, was used as an electrification material and as a matrix of a CNT-filled composite electrode, giving TENG the ability to instantly heal from mechanical damage (almost completely restored by 3 min without additional stimuli) [215].

Biocompatible ionic gels with shape memory are made from (3-acrylamidophenyl) boronic acid, acrylamide, and chitosan containing catechol groups (Figure 48) [216]. The reversible cross-linkers of H-bonding and dynamic covalent bonds allow the fast self-healing of gels in a matter of minutes, but also with a large stretchability (e.g., 12–200 times the tensile length) and plasticity for shape memory on irregular surfaces.

SHPs, which showed good healing properties, were obtained by DA and urethane formation reactions [217]. The healed samples maintained at least 50% of their initial strength. Polymeric elastomers with integrated stretchable and self-healable characteristics have been developed by cross-linking hyperbranched polymers with flexible segments [218]. Due to the reversibility of the imine and disulfide bonds used, the elastomers have shown good self-healing properties, and the healing efficiency reached 99% under ambient conditions.

PU was synthesized using 2,6-diamimopyridine and cystamine as chain extenders that were subsequently complexed with Zn^2+^ ions (Figure 49a) [219]. It was found that pyridine fragments, together with the formation of M–L bonds, significantly affect the microphase separation by interfering with the H-bond. Moreover, its self-healing efficiency is simultaneously improved to 96.64 ± 1.52%. The electrode film resulting from the incorporation of Ag NWs into this polymer exhibits self-healing properties by restoring the conducting network using a self-healable matrix. SHPs were prepared based on synergistic multiple non-covalent bonds between carboxymethyl guar gum, PAA and ferric metal ions (Fe^3+^) in a covalent polymer network (Figure 49b) [220]. The reversible and dynamic nature of multiple M–L interactions explains the good self-healing capabilities and high self-healing efficiency.

A simple method for producing healable elastomeric vitrimers has been demonstrated by creating dynamic dual cross-links of boronic esters and M–L bonds in commercial rubber (Figure 50) [221]. The covalently cross-linked networks are able to change the topology through the transesterification of boronic ester, thereby imparting their healing ability. In particular, the mechanical properties can be significantly improved by introducing sacrificial M–L bonds into the network without compromising the healing or reprocessing ability.

Of interest is the self-healing polydiacetylene–PAA–Cr^3+^ hydrogel (Figure 51a) [222]. Cr(H_2_O)_6_^3+^ complexes act as cross-linkers, maintaining the stability of the hydrogel framework through electrostatic binding to carboxylate moieties in both PAA and polydiacetylene, further contributing to the inclusion of a high concentration of water molecules necessary to maintain the elasticity of the hydrogel. The process of obtaining such hydrogels includes the complexation of PAM with PAA chelated with Fe^3+^ ions, shape processing, and the loading of NaCl salts (Figure 51b) [223]. The formation of hydrogels occurs due to the formation of M–L bonds between PAA and Fe^3+^ ions and H-bonds between PAM and PAA. Most likely, loading with NaCl leads to the shrinkage of polymer chains and the formation of hydrophobic interactions in hydrogels.

SHP was obtained by blending Fe^3+^ ions with polyvinyl alcohol acetoacetate/PAM hydrogel to form a DN hydrogel that combined M–L coordination and physical crosslinking in one system (Figure 52a) [224]. The M–L coordination of the Fe^3+^/PVAA network provided self-healing ability. The resulting hydrogel exhibited a high healing efficiency (80% within 24 h) and excellent and repeatable stretch and compression rebound characteristics. In addition, hybrid network hydrogels show rapid self-healing with ionic conductance (within 0.36 s) at room temperature without any other external stimuli. Elastomers have been synthesized containing hard and soft segments. The former is based on double-locked nanodomains due to hydrophobic interactions and iron–catechol complexation. The latter include an inter-nanodomain flexible polymer matrix, with a high fracture energy of 24,000 J/m^−2^ (Figure 52b) [225]. The main characteristics of the self-healing process include the following parameters: room temperature, 1 h, and recovered adhesion energy of 700 J/m^−2^ between the self-sealed films.

A design strategy has been proposed whereby both low crosslink density and high dynamic crosslink bonds are introduced into the polyurea network to achieve the tough and low-temperature SHP of PPG-PDMS-Zn [226]. The dynamic characteristics and self-healing ability of SHP were effectively improved via the combination of the unique feature of the dynamic exchange of weak Zn-urethane coordination bonds and the low-temperature effect of the inhibition-dissociation of H-bonds. As a result, SHP exhibits a high self-healing efficiency of 97% at −20 °C. The PPG segment has been introduced into the PDMS backbone because it can provide both H-bonding and Zn-urethane bonding sites. A self-healing PU has been designed using a dual-dynamic network design in which multiple H-bonds and M–L bonds (zinc–imidazole interaction) have been incorporated into the PU matrix (Figure 53) [227]. These metallopolymers demonstrate good self-healing abilities. First, isophoronediisocyanate is used as a hard segment that resists crystallization, while flexible polytetrahydrofuran is used as a soft segment. These two parts improve the mobility of the chain, which is beneficial for the self-healing process. Second, adipicdihydrazide, which endows the polymer with multiple H-bonds, was chosen as the chain extender. Third, *N*-(3-aminopropyl)-imidazole acts as an end-capping reagent that can coordinate with Zn^2+^, creating a dynamic M–L bond. In general, a dual-dynamic cross-linked system can simultaneously improve the mechanical properties and self-healing ability of the materials at room temperature.

A simple method is proposed for the preparation of a Silly Putty-type hydrogel with good self-healing, based on dynamic M–L interactions and H-bonds, by simply mixing PVA phytate, pyrrole and Fe^3+^ [228]. An efficient silicone-based SHP was developed from supramolecular silicone polymers and nanoparticles of silicon dioxide, which can be sprayed onto solid substrates and assembled into porous films with increased water repellency [229]. The abundant H-bonds and M–L bonds in SHPs provide the substrate, with additional binding and damage-healing properties. A universal electrode is reported that fully integrates a three-dimensional electrode material, a self-healing hydrogel, and an electrolyte [230]. A three-dimensional porous carbon sponge (CS) serves as both an electroactive material and a carrier for the Fe^3+^-cross-linked sodium polyacrylate–LiCl hydrogel electrolyte. The omnidirectionally integrated electrode exhibits excellent self-healing characteristics; moreover, its capacity retention rate remains high (91.6%) even after five cutting/healing cycles. SHPs were prepared based on ultra-long Ag NWs consisting of binary-networked hydrogels [231]. The flexible hydrogels can provide a high healing efficiency (94.3%), which is explained by the strong covalent bond and reversible physical interaction of the structured binary-network. In a three-dimensional hydrogel, the physical interactions of H-bond and M–L coordination are largely reversible, and the polymer matrix can be quickly reconstituted after reconnection, providing highly efficient mechanical recovery. However, the hydrogel shows a reduced self-healing efficiency under repeated fractures. SHP was prepared by introducing PDA nanoparticles and water-soluble ILs into the hydrophobic association PAM [232]. The catechol groups from PDA have played an important role during the exfoliation process based on the H-bond, the hydrophobic interaction and metal complexation. TENG has been demonstrated as not only an energy source, but also as a self-powered e-skin (Figure 54) [40]. It consists of a metallopolymer as a triboelectrically charged layer and H-bonded ion gel as an electrode. In this TENG, both the triboelectrically charged layer and the TENG electrode are inherently and autonomously self-healing under environmental conditions. In addition, it has a fast healing time (30 min, 100% efficiency at 900% strain). Even after 500 cutting and healing cycles or at a maximum stress of 900%, TENG retains its functionality. This TENG has a number of significant advantages: fast self-healing under environmental conditions, high transparency, internal stretchability, and the abilities of energy-extraction and active-sensing. This allows it to be applied in many areas, from deformable, portable or transparent electronics, to smart interfaces and artificial skins. In addition, TENG is also promising as a self-powered tactile-sensitive skin in a variety of human–machine interfaces (HMI), including smart glass, epidermal controllers and phone panels. It is important to emphasize that the generated electricity can either be used directly or stored to power commercial electronics.

A method for creating a dual ionic network for natural rubber-based carboxylate ionomers with a high self-healing efficiency was demonstrated [233]. Maleic anhydride was grafted with rubber using the anionic mechanism to create the first ionic network with n-butyllithium as a metallization reagent. Subsequently, zinc dimethacrylate reacted with the maleic anhydride-modified rubber, and a second ionic network was built. The dual ionic network showed a high self-healing efficiency of 75% in the full-cut mode. SHP (Figure 55) with an excellent resistance to low temperatures has been demonstrated, in which both the M–L bonds and tetrahedral borate interactions in binary-networked frameworks are responsible for the high healing efficiency (~90.4%) [234]. The low temperature (−25 °C) tolerance of SHP is favorable for all-weather applications.

A self-healing collagen-based hydrogel has been developed based on dynamic network chemistry, consisting of dynamic imine linkages between collagen and dialdehyde guar gum, as well as diol-borate ester bonds between guar gum and borax (Figure 56a) [235]. The as-prepared collagen-based hydrogels showed a good injectability and rapid self-healing capacity (within 3 min). A PVA/cellulose nanofibril hydrogel with dual-cross-linked networks was synthesized for self-healing sensors (Figure 56b) [236]. The presence of dynamic borate bonds, metal–carboxylate coordination bonds and H-bonds determines the self-healing ability of the hydrogel for 15 s without any external stimuli compared to traditional PVA hydrogels.

Noteworthy is waterborne PU, which undergoes self-healing in air at body temperature or in water under the influence of ultrasound [237]. Importantly, it can be re-healed over three cut–heal cycles at the same fracture site while maintaining the rate of recovery of tensile strength and elongation at break. In addition, its self-healing ability can be improved by introducing aromatic disulfides along with multiple hydrogen and ionic bonds into the PU network. In this case, the modified PU film has a tensile strength of 18.4 MPa and an elongation at break of 1260%, which is higher than an unmodified PU film at room temperature. A PU-based material was developed with a high self-healing efficiency at 80 °C on the basis of reversible DA bonds as well as a zinc–ligand structure (Figure 57a) [238]. After self-healing, the tensile strength was 25.85 MPa, leading to the high self-healing efficiency of 90.8%. In addition, by introducing carbonyl iron powder, PU-containing carbonyl iron powder can be achieved, which exhibits a microwave-assisted self-healing property, and a self-healing efficiency of 92.6% can be reached in 3 min. A hydrogel with fast self-recovery is constructed (Figure 57b) [239]. The proposed double network hydrogel is achieved by combining the chemically cross-linked PAM and physically cross-linked gelatin network, followed by sodium citrate solution immersion. In order to improve the recovery properties, abundant physical interactions are introduced to the hydrogel as reversible sacrificial bonds, including ionic crosslinking, H-bond, and hydrophobic associations. These reversible physical interactions enable the hydrogel to reconstruct interactions quickly during stretching, resulting in a fast self-recovery.

A self-healing (90.8%) binary network hydrogel PAA-PANI is proposed [240]. Metal ion (Fe^3+^)-coordination, strong H-bonds and electrostatic interactions provide satisfactory self-healing properties. First, trivalent Fe^3+^ can chelate the carboxyl groups of PAA, and reversible tridentate coordination promotes its ability of self-healing. Second, amino groups distributed in PANI can create H-bonds between adjacent chains, which helps to form a strong “dynamic zipper”. This adheres tightly without deformation and quickly self-heals after rupture. Third, strong electrostatic interactions between PAA and PANI improve the mechanical properties. A toughening strategy was proposed for SHP by introducing ionic cluster complexes of iron with carboxylate (Figure 58) [241]. The resulting SHP simultaneously exhibits a tough mechanical strength, a high stretchability, a self-healing ability, and processability at room temperature. The superior performance of these SHPs is attributed to the hierarchical existence of four types of dynamic combinations in a high-density dry network, including dynamic covalent disulfide bonds, non-covalent H-bonds, iron–carboxylate complexes, and ionic cluster interactions.

A PU has been developed containing triple synergistic dynamic bonds that exhibits excellent self-healing characteristics (Figure 59a) [242]. Disulfide bonds, and reversible boronic ester bonds located at the cross-linked point and backbone, are sequentially broken up when damaged, which promotes better chain fluidity and endows the material with a superior healing efficiency and multiple healing cycles. A PU elastomer, based on a Cu(II)–dimethylglyoxime–urethane complex with synergistic triple dynamic bonds, has been developed (Figure 59b) [243]. The elastomer demonstrates the good mechanical characteristics of self-healing elastomers at room temperature, with a tensile strength and toughness up to 14.8 MPa and 87.0 MJ m^−3^, respectively. The coordination of Cu(II) plays a critical role in accelerating the reversible dissociation of dimethylglyoxime–urethane, which is important for the superior performance of the self-healing elastomer.

## 5. SHPs with Shape Memory

In recent years, polymers in which self-healing is accompanied by a shape memory effect have attracted considerable attention from researchers [46,91,244,245,246,247,248]. In general, shape memory polymers (SMPs) are polymeric materials that can change from a permanent shape to one or more temporary shapes, and subsequently recover their permanent shape in response to external stimuli (for example, light, magnetism, heat, or chemicals) [249,250,251]. SMPs are currently the subject of extensive research due to their promising applications in biomedicine, textiles, aerospace, etc. [252,253]. As a typical example, we note the tunable microarchitectured shape memory TENGs that exhibit self-healing abilities in both macro shape and micromorphology, while providing improved and variable triboelectric output (~150–320 V, ~2.5–4 μA cm^−2^) due to the increased frictional effect enabled by the high surface roughness (Figure 60a) [254]. At the micro level, the self-healing ability caused by thermal stimuli makes the deformed mats capable of rebuilding their original microstructures, providing durable TENGs with extended service lives. With cellulose oleoyl ester, waterproof mat-based TENGs can be made with a stable rough surface for harvesting energy from both cold and hot water. A deformed waterproof TENG is found to be recoverable in shape under hot water. The gradient surface roughness provides discernible triboelectric outputs during the structural recovery process, allowing the design of a water energy harvester with a sensing ability for water temperature (25 ± 5 °C to 95 °C), which is promising for self-powered, waterproof, wearable electronics and smart wastewater management systems. In another interesting example, a healing and shape memory dual-functional polymer with improved mechanical properties and stimuli responses was developed, which was used to create a TENG with superior reliability and versatility (Figure 60b) [255]. It is important that the short-circuit current and the open circuit voltage of the healed device can retain their original values without explicit changes. In addition, thanks to the TENG’s shape memory ability, the device can be used not only as a smart collective insole for flatfoot treatment and gait analysis, but also as a self-powered fire alarm and escape indicator system, due to its temperature-sensitive properties.

A self-healing TENG has been achieved with SMP, in which performance can be restored after morphological damage caused by high working force (Figure 61a) [256]. PU was used for this purpose because it is easy to use, has a fairly low healing temperature of up to 55 °C, and allows the TENG to heal without damaging its other components. After heating above the T_g_ of PU for 30 s, the resulting TENGs were able to recover their original micro-patterned structure and mechanical harvesting capacity (Figure 61b). However, the SMP’s ability to recover its shape has a limit. In particular, TENG was healed more than 30 times consecutively after compression with a strong force of 12 kgf. Under excessive force, the SMP can reach its limit and be so severely degraded that recovery will be impossible. In addition, the SMP has a limited lifespan, and can gradually degrade over time. Hence TENGs do not last forever, but the stress that they can bear is higher than most common polymers used in TENG.

The combination of DA chemistry and ionic interactions makes it possible to obtain dynamic cross-linked networks with high shape memory indices, especially at higher temperatures [257]. Of interest is SMP, in which the polysiloxane network is cross-linked by dynamic DA bonds [258]. The self-healing process of polysiloxane involves solid–liquid–solid transformation. It should also be noted that a self-healing nanocomposite with a high tensile stress and excellent electromechanical characteristics was obtained by introducing graphene nanosheets into a polysiloxane elastomer.

TENG using SMPs is demonstrated (Figure 62) [259]. The device is based on the mechanism of a flexible TENG using the thermally triggered shape transformation of organic materials to efficiently harvest mechanical energy. An SMP was synthesized by incorporating a semicrystalline thermoplastic polymer into a chemically cross-linked elastomer, forming a semi-interpenetrating polymer network wherein semicrystalline thermoplastic polymer chains were embedded in the elastomer network. When the SMP was deformed at a temperature above the melting point of PCL, the fine crystals melted and deformed along with the elastomer network. After cooling without removing the load, the PCL chains reformed a physically cross-linked network through small crystals that could lock the deformed network. Importantly, the shape memory property makes it different from all other deformable TENGs, as it can adapt to almost any surface configuration through a thermal process and maintain its shape without any external retention. In combination with a conductive liquid, TENG can change its color as well as its shape, which can significantly expand its range of applications. A pressure-sensitive electrical output within a defined range allows the TENG to be a mechanical sensor. Multifunctional device designs based on other new polymer materials that have unique properties, such as photosensitivity and self-healing, should find widespread use in soft robotics, medical devices and artificial skin, through integration with energy storage devices to ensure sustainable energy supply.

The examples of such thermal-induced SMPs are numerous [260,261,262,263,264]. We should also note a hydrogel with shape memory and self-healing properties, which was obtained by mixing phenylboronic acid-modified gelatin, catechol-modified carboxymethyl chitosan, 3,5-dinitrosalicylic acid, and Eu^3+^ ions [265]. The remarkable shape memory of the resulting hydrogel was realized. In addition, a self-healing ability was also achieved through the dynamic bonds in the hydrogels. A thermoresponsive SHP thermoset, based on the DA reaction between furan and maleimide groups, has been developed [266]. The polymer exhibits a glass temperature-induced shape memory effect, which brings the crack faces into intimate contact, allowing the DA bridges located in the crack surfaces to be reformed. In addition, the inclusion of PDMS segments and excess maleimide functional groups in the network, in addition to DA bridges, promotes the mobility of dynamic groups and the healing reaction, thereby increasing the ability of the material to heal damage at moderate temperatures. The healing of scratches was made possible by a one-step heating process at a moderate temperature of 60 °C for 4 h, while the recovery of tensile strength reached more than 85%. Dual-physical cross-linked network elastomers with shape memory have been developed by introducing dynamic non-covalent bonds to enhance self-healing [54]. H-bonds and hydrophobic associations play an important role in improving the self-healing ability of elastomers. The healing efficiency reaches 95.32% after 3 h at room temperature without any external stimuli. The introduction of dynamic non-covalent bonds into the elastomers leads to excellent shape memory; in particular, the maximum shape recovery efficiency is 72.02 ± 6.94% after 10 s at 37 °C, and full shape recovery is achieved after 39.67 ± 1.25 s. The main contribution to highly efficient self-healing is made by hydrophobic interactions between the benzene ring groups.

Of interest are the SMPs obtained by blending furan-terminated thermoplastic polyurea-urethanes with dynamic disulfide bonds and epoxy oligomers with dangling furan groups [267]. The DA reaction between the maleimide groups in bis(4-maleimidophenyl)methane and the furan groups in epoxy oligomers results in the phase separation of epoxy domains, leading to improved shape memory properties. For example, fixation ratios of 91 and 99% are obtained by cold drawing or heating, respectively. The healing efficiency is 80%, and can be improved by blending miscible epoxy oligomers (Figure 63) [268]. Interesting results were obtained in the presence of bis(4-maleimidophenyl)methane. In particular, the DA reaction between furan groups and maleimide groups in bis(4-maleimidophenyl)methane resulted in the separation of the cross-linked epoxy domains phase from the PU matrix to form microscale domains. The result was impaired self-healing and easier shape fixation.

Hindered urea bonds have been used for reversible dynamic covalent cross-linking to create hindered thermosetting polyureas [269]. The resulting polymers exhibit high permanent reshaping and self-healing capabilities. In particular, the combination of conventional elastic shape memory with the solid-state plastic shape reconfiguration in these polyureas expands the geometrically complex shapes of SMPs. It is important to note that the introduction of longer and more bulky substituents into the polyureas can simultaneously endow polymers with superior mechanical properties and many important functions.

A self-healing composite based on a vitrimer should be noted, which was used to close cracks and provide a certain reinforcement of the composites [270]. All the composites can heal, to some extent, a 1 mm crack produced by tension. It is important that the healing efficiency continuously increases from 41.1 to 58.6% from the second to the fifth fracture. Apparently, these properties of the vitrimer are due to a number of factors, i.e., gradual compaction of the PCL-diol adhesive layer, the improvement of interfacial binding and physical entanglement, and a possible transesterification reaction between the PCL healing agent and the vitrimer matrix.

## 6. Summary and Future Outlook

TENGs represent an advanced system for harvesting mechanical energy, which has allowed them to demonstrate high vitality and enormous advantages, creating great prospects for their use in various fields of human activity. The inclusion of SHPs in the design and the development of TENGs is a promising strategy for ensuring the synergy of the reliability and durability of such devices. As the content of this review shows, the current stage in the development of SHPs for TENGs has reached its peak in the accumulation of experimental facts, their theoretical interpretation, and their generalization. In addition, an increasing number of research groups are being drawn into this promising area of modern polymer chemistry. One of the latest advances is the use of SHPs as components for hybrid devices, including TENGs, electromagnetic generators and piezoelectric nanogenerators. This approach is revolutionary, but so far, its further development largely depends on challenges and issues that have not yet been fully resolved. In particular, while there have been very promising advances to date, there is still a long way to go before the introduction of self-healing wearable TENGs in real-world or commercial applications.

How do we ensure the development of this interesting and promising direction of polymers for advanced technologies?

First of all, it is important to emphasize that the range of SHPs used for these purposes is huge, and is not limited to the examples considered here. The achievements made in the modern synthetic chemistry of polymers provide a wide range of SHPs with a large arsenal of design strategies for their manufacture and scalability, and ensure an acceptable price/quality ratio, which creates a methodological platform for obtaining promising SHPs for TENGs. Therefore, it is safe to say that the development of such polymers will continue at a high level. For some special application scenarios, it is necessary to develop and synthesize new SHPs to create high-performance TENGs, such as energy harvesting in liquid conditions, in vivo, and even in restricted access environments.

Despite the huge potential of SHPs, their commercialization, like TENGs in general, is still far from being a reality. It is important to note that most of the stretchable triboelectric SHPs are based on commercial elastomers. However, SHPs are generally more expensive than commercial polymers, mainly because they require more synthetic steps and chemical modification processes. In fact, there are important problems that need to be overcome in order to start the effective technological development of these systems. In addition, self-healing TENGs usually undergo additional processing to ensure reliability and stability [127,271,272,273]. In particular, this is due to the fact that the processes of their self-healing proceed at high temperatures (90 °С) for hours [274]. The cracked triboelectric layer does not significantly affect the final electrical performance; however, the electrical output of TENG drops sharply when the electrode layer is damaged or broken [183], and hence the self-healing electrode layer is substantial and significant. A healing TENG with a tensile load of ≈4% was obtained through the external layer to accelerate healing; however, the carbon-based electrode was opaque, and the device was recovered by heating to 130 °C. The development of TENGs in which both the triboelectric layer and the electrode autonomously self-heal under ambient conditions, and at the same time have high transparency and high stretchability, is an important step for deformable/wearable/flexible energy sources [24,42,275,276]; obtaining such TENGs is extremely difficult. Among the few examples of such systems, we note the self-healing TENG, made by combining a temperature-responsive PCL with flexible Ag NWs, in which both friction surfaces and conductive layers can self-repair [277]. In particular, when the top surface of the electrode is damaged, the PCL-based SHP will intenerate by heating, and flow towards the wound to achieve the goal of self-healing. If the conductive layer in the TENG is also damaged, PCL will also cause the Ag NW network at the bottom of the electrode to move to heal during the heating process.

The mechanical stability, stretchability and power density of SHPs are far from practical applicability. Thus, there is a need to develop triboelectric SHPs with excellent stretchability and high mechanical properties. In addition, most SHPs are not transparent, or require external stimuli (heat, moisture, light, etc.) to trigger the healing process. Therefore, SHPs must be improved to achieve better mechanical properties and stretchability. However, achieving extreme stretchability (with good energy-harvesting performance) and healing (full recovery of its performance after mechanical damage) is still a difficult challenge [278].

An important problem is the development of a standard procedure for characterizing SHPs for TENGs, since each sample is tested in its own way, and therefore it is difficult to compare different types of SHPs. Unfortunately, for many SHPs, the working mechanism of action and behavior in TENGs is not fully understood. Additional problems arise from the requirements for the precise control and start-up of such systems. Importantly, most SHPs require external stimuli (heat, moisture, light, etc.) to trigger the healing process. Therefore, in the future, it is necessary to improve the SHP with a spontaneous healing process.

In addition, an important task is to find a correlation between the composition, spatial organization and properties of SHPs, which greatly complicates the scientifically grounded approach to structuring these systems and predicting their application in TENGs. It should be emphasized that the main trend in the coming years will be towards the development of multifunctional SHPs. Advances in these areas could provide significant advantages in terms of strength and resistance to damage for various SHPs, allowing the use of new generations of human-compatible electronics.

In the near future, new innovations are expected to appear in SHPs for TENGs. To this end, SHPs must be designed with three main characteristics—fast healing, biocompatibility and cost-effectiveness—so that self-healing TENGs can be seen in real life or on the market in a short time.

## Figures and Tables

**Figure 1 polymers-12-02594-f001:**
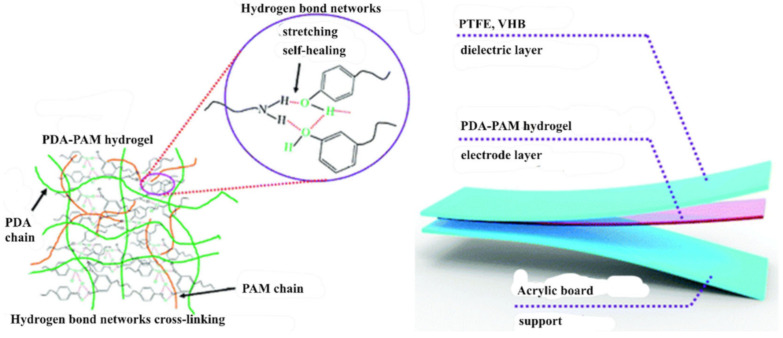
The mechanism of the hydrogel’s self-healing properties and design of TENG. Reproduced from [104] with permission from The Royal Society of Chemistry.

**Figure 2 polymers-12-02594-f002:**
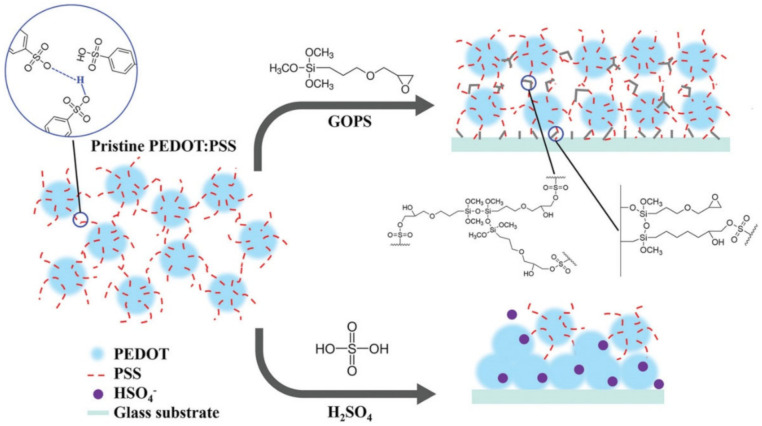
Schematic diagram of proposed PEDOT:PSS molecular structure changes via crosslinker addition and acid post-treatment. Reproduced from [111] with permission from John Wiley and Sons.

**Figure 3 polymers-12-02594-f003:**
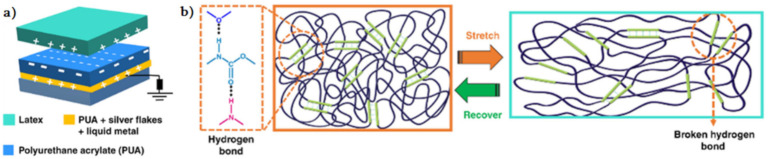
(**a**) Schematic diagram of the complete stack of stretchable and healable TENG and (**b**) a schematic representation of the breaking and reforming of H-bond resulting in high stretchability. Reprinted from [113].

**Figure 4 polymers-12-02594-f004:**
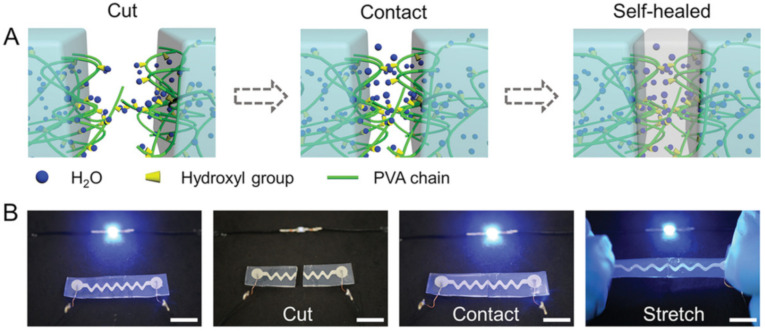
Self-healing of the LM-based hydrogel electronics. (**A**) Schematic illustration of the self-healing process of the PVA hydrogel. (**B**) Photographs of the LM–hydrogel device before and after cutting. The self-healed device under stretching is also shown. Scale bars: 1 cm. Reproduced from [114] with permission from John Wiley and Sons.

**Figure 5 polymers-12-02594-f005:**
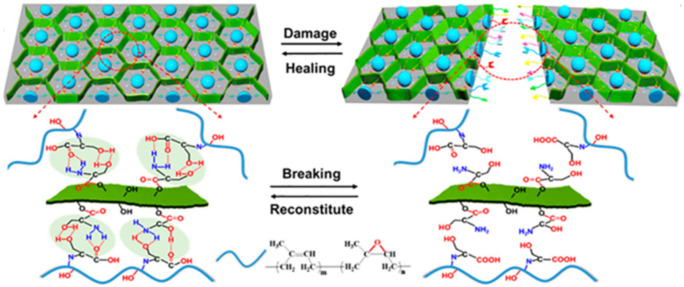
Dynamic breakage and reconstitution of interfacial supramolecular H-bonds in supramolecular elastomer. Reprinted with permission from [115]. Copyright (2020) American Chemical Society.

**Figure 6 polymers-12-02594-f006:**
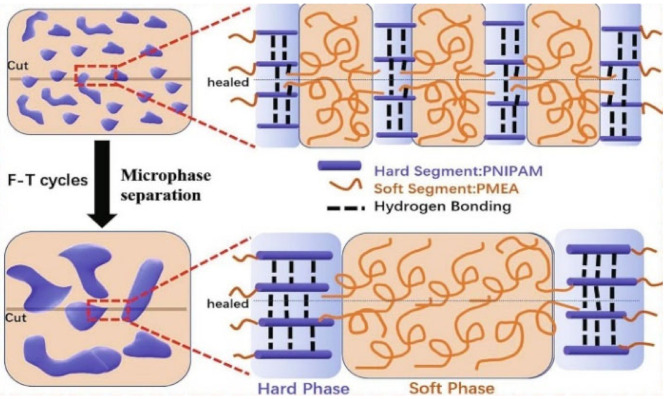
The proposed self-healing mechanism based on the reversible H-bonds and microphase separation structures. Reproduced from [122] with permission from Elsevier.

**Figure 7 polymers-12-02594-f007:**
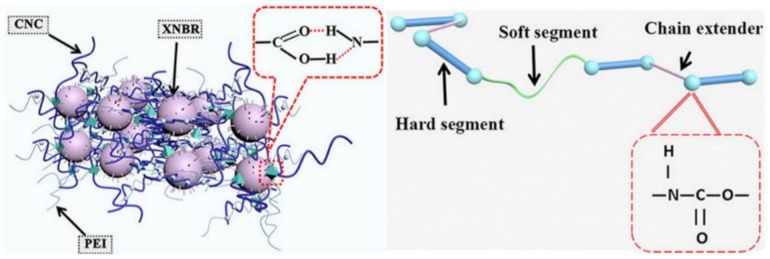
Self-healing mechanisms of composites and PU, respectively, where CNC, XNBR and PEI are cellulose nanocrystals, carboxylated nitrile rubber and polyethylenimine, respectively. Reprinted with permission from [123]. Copyright (2020) American Chemical Society.

**Figure 8 polymers-12-02594-f008:**
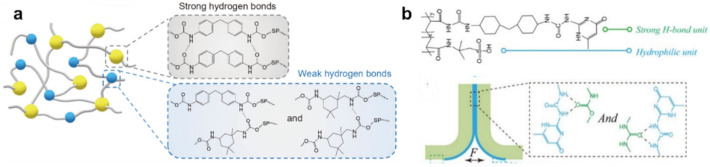
(**a**) Chemical structure of PDMS elastomer and associated conductive morphology diagram. Reproduced from [125] with permission from Springer Nature. (**b**) Scheme of TENG and H-bond at the interface. Reprinted with permission from [126]. Copyright (2020) American Chemical Society.

**Figure 9 polymers-12-02594-f009:**
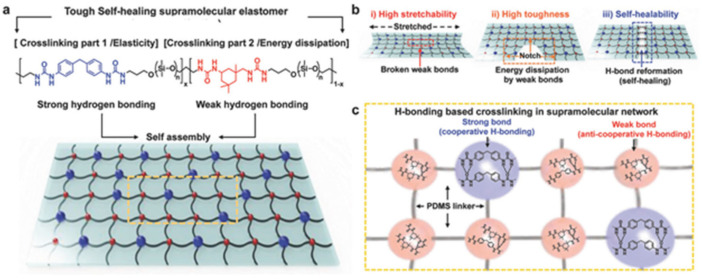
(**a**) Chemical structure of the elastomer and proposed ideal supramolecular structure. (**b**) Diagrams of stretched polymer film (left), notched film (middle), and healed film (right). (**c**) Possible combinations of H-bonds for strong and weak bonds, respectively. Reproduced from [127] with permission from John Wiley and Sons.

**Figure 10 polymers-12-02594-f010:**
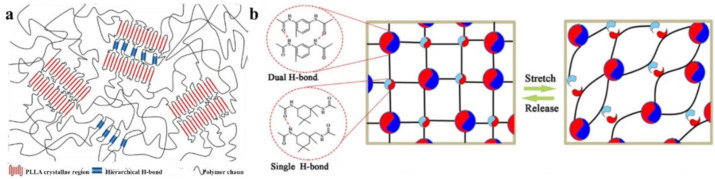
(**a**) Schematic representation of SHPs based on poly(l-lactic acid) (PLLA) crystalline segments and hierarchical H-bond. Reprinted with permission from [128]. Copyright (2020) American Chemical Society. (**b**) Diagram image of the networks with H-bonds of different strength. Reproduced from [129] with permission from Elsevier.

**Figure 11 polymers-12-02594-f011:**
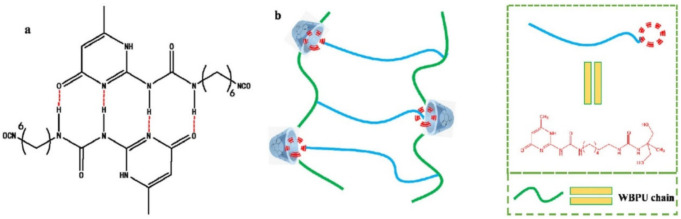
(**a**) Illustration of the self-healing process and the dynamic of the hierarchical H-bond interaction. (**b**) Schematic diagram of the interaction between guest and host. Reproduced from [131] with permission from Elsevier.

**Figure 12 polymers-12-02594-f012:**
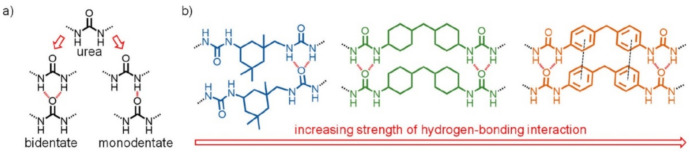
(**a**) Various H-bond interactions in ureas. (**b**) H-bond interactions in the PDMS-based poly(urea)s depending on their strength. Reprinted with permission from [135]. Copyright (2020) American Chemical Society.

**Figure 13 polymers-12-02594-f013:**
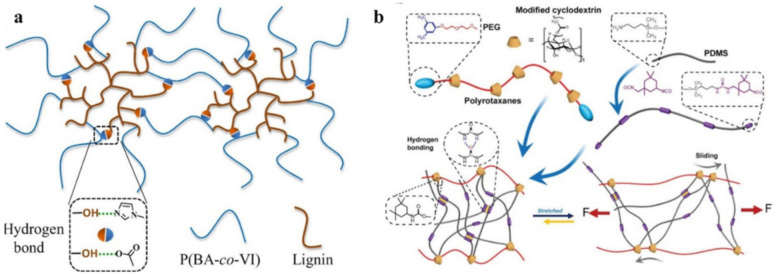
(**a**) Proposed H-bond interaction in the elastomers. Reproduced from [136] with permission from Elsevier. (**b**) Schematic representation of an elastomer cross-linked with polyrotaxanes and H-bond, illustrating the sliding behavior of cyclodextrins when the elastomer was stretched. Reproduced from [137] with permission from John Wiley and Sons.

**Figure 14 polymers-12-02594-f014:**
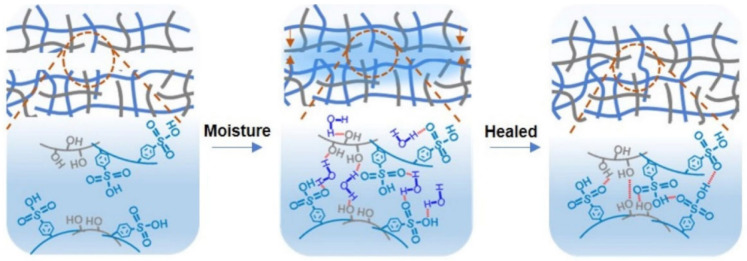
Scheme of the moisture-induced healing process among the PSS-PVA chains: an initial crack (left), its fusion under the influence of moisture (middle) and restoration to its original state (right). Reproduced from [140] with permission from Elsevier.

**Figure 15 polymers-12-02594-f015:**
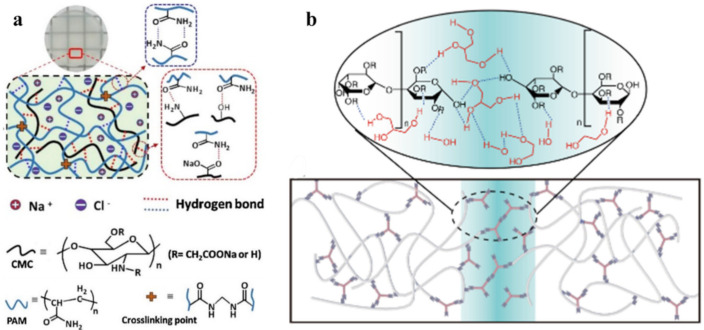
(**a**) Composite hydrogels schematic. Reproduced from [144] with permission from Elsevier. (**b**) Schematic diagrams of the self-healing properties of an elastomeric gel. The gray lines represent hydroxyethylcellulose polymer chains and form cross-linked networks. The purple shapes on polymer chains and brown short lines form primary H-bonds between polymer chains and glycerol. Reproduced from [145].

**Figure 16 polymers-12-02594-f016:**

The proposed mechanism of healing for the as-prepared PDMS-PU film. Reproduced from [45] with permission from Elsevier.

**Figure 17 polymers-12-02594-f017:**
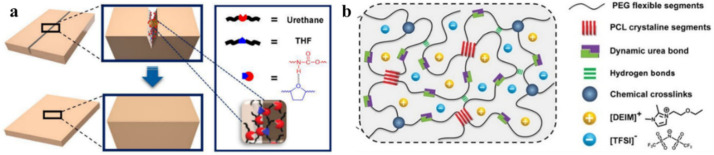
(**a**) Schematic representation of the PU’s self-healing mechanism. Red and blue represent a urethane group and a heterocyclic group, respectively. Reproduced from [146]. (**b**) Schematic structure of the PU-IL_2_ ionogel. Reproduced from [147] with permission from John Wiley and Sons.

**Figure 18 polymers-12-02594-f018:**
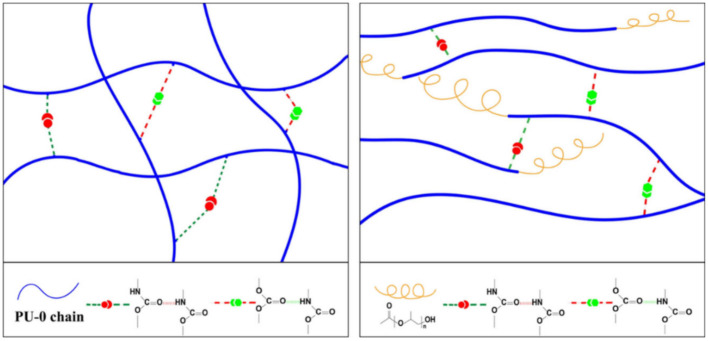
Schematic model of H-bonds in PU (**left**) and PU-containing PPG segments (**right**). Reproduced from [148] with permission from Springer Nature.

**Figure 19 polymers-12-02594-f019:**
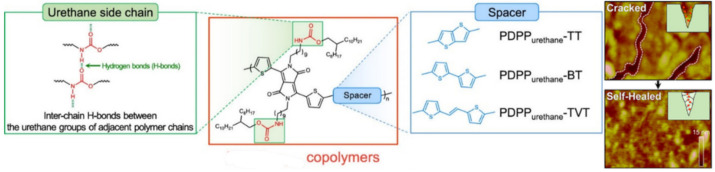
Molecular structures of the copolymers and images of cracked and self-healed samples. Reprinted with permission from [149]. Copyright (2020) American Chemical Society.

**Figure 20 polymers-12-02594-f020:**
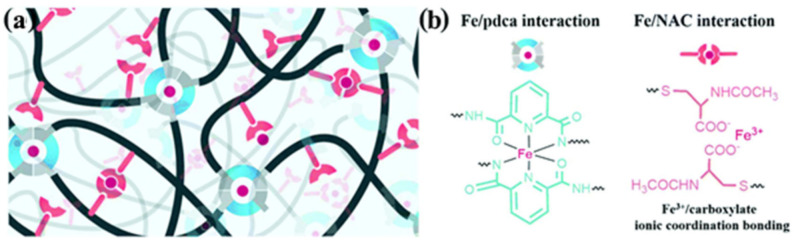
(**a**) Schematic representation of a dual-cross-linked PDMS network. (**b**) Illustration of the structures of the Fe/pdca complex (blue) and Fe/NAC complex (red), where pdca and NAC are pyridinedicarboxamide and *N*-acetyl-l-cysteine, respectively. Reproduced from [150] with permission from The Royal Society of Chemistry.

**Figure 21 polymers-12-02594-f021:**
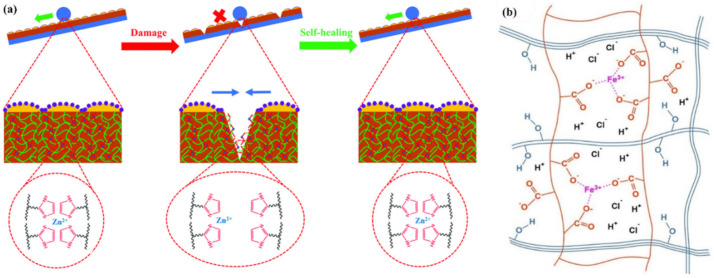
(**a**) Elastomer self-healing through bond reforming. Reproduced from [152] with permission from The Royal Society of Chemistry. (**b**) Completed molecular structures of Fe^3+^/PAA-PVA hydrogels. Reproduced from [153] with permission from Elsevier.

**Figure 22 polymers-12-02594-f022:**
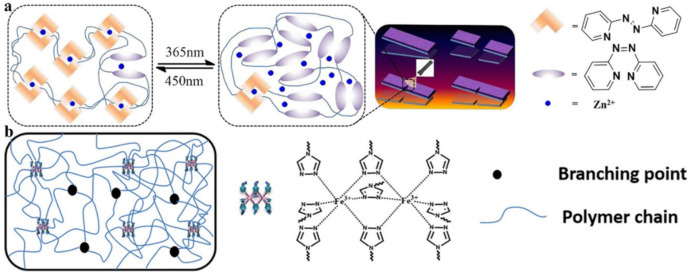
(**a**) Schematic diagram of the light-healing process. Reproduced from [154] with permission from Elsevier. (**b**) Preparation of SHP with Fe–triazole interactions. Reproduced from [155] with permission from John Wiley and Sons.

**Figure 23 polymers-12-02594-f023:**
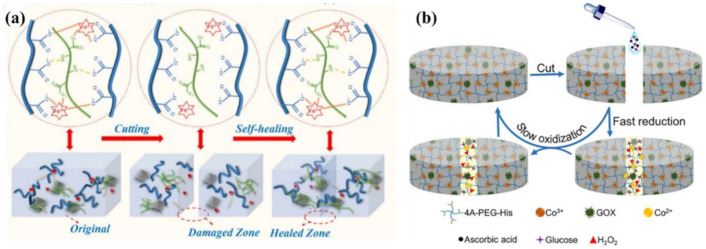
(**a**) Schematic representation of the self-healing mechanism of hydrogels. Reproduced from [156] with permission from Elsevier. (**b**) Schematic of glucose oxidase (GOX)-regulated transient healing of cobalt cross-linked polymer networks. Reprinted with permission from [157]. Copyright (2020) American Chemical Society.

**Figure 24 polymers-12-02594-f024:**
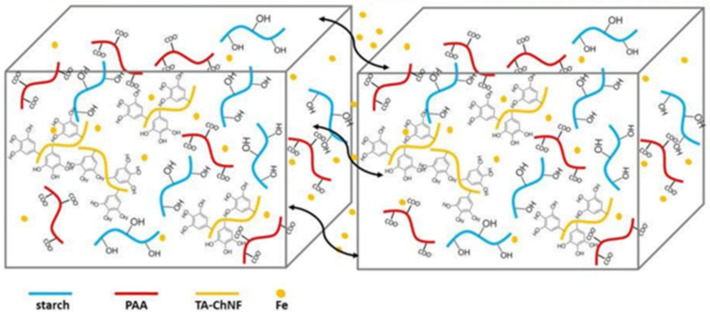
Hydrogel electrical self-healing mechanism. Reprinted with permission from [160].

**Figure 25 polymers-12-02594-f025:**
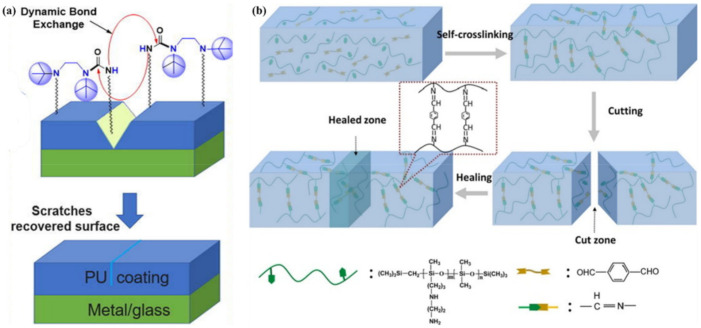
(**a**) Chemistry of the dynamic behavior of the bulky urea bond. Reproduced from [161] with permission from Elsevier. (**b**) Schematic diagram of PDMS synthesis and self-healing process. Reproduced from [162] with permission from Elsevier.

**Figure 26 polymers-12-02594-f026:**
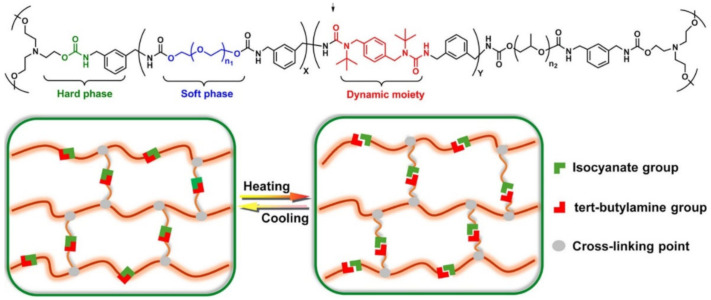
Diagram of PU networks and their self-healing mechanism. Reprinted with permission from [165]. Copyright (2020) American Chemical Society.

**Figure 27 polymers-12-02594-f027:**
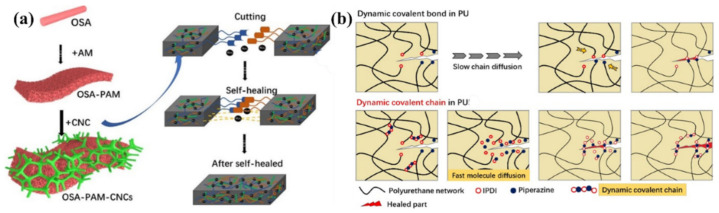
(**a**) Schematic diagram of the preparation of OSA-PAM-CNCs hydrogel and the mechanism of its self-healing, where OSA is an oxidized alginate. Reprinted from [168]. (**b**) Difference between the healing processes of PUs with dynamic covalent bond or dynamic covalent chain. Reproduced from [169] with permission from Elsevier.

**Figure 28 polymers-12-02594-f028:**
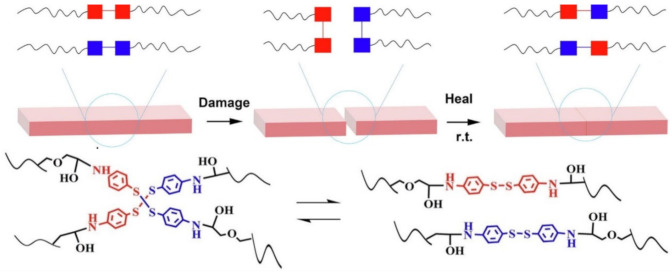
The self-healing siloxane elastomer. Reproduced from [173] with permission from Elsevier.

**Figure 29 polymers-12-02594-f029:**
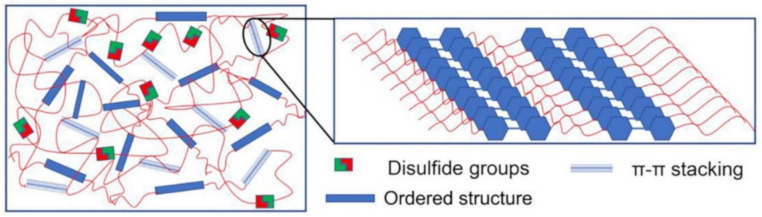
Schematic diagram of the structure of a self-healing and reproducible disulfide-based liquid crystal elastomer. Reproduced from [174] with permission from John Wiley and Sons.

**Figure 30 polymers-12-02594-f030:**
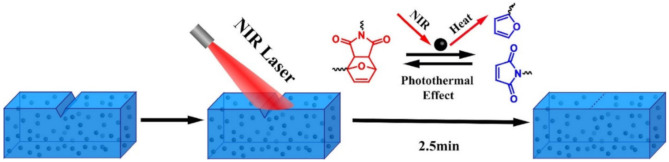
SHP self-healing process triggered by NIR laser. Reproduced from [176] with permission from John Wiley and Sons.

**Figure 31 polymers-12-02594-f031:**
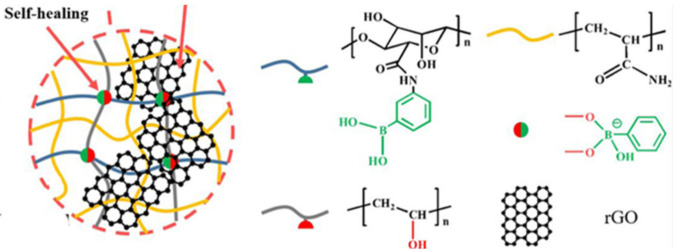
Schematic representation of the fabrication of SHP assembled from a conductive non-drying nanocomposite organohydrogel. Reprinted with permission from [182]. Copyright (2019) American Chemical Society.

**Figure 32 polymers-12-02594-f032:**
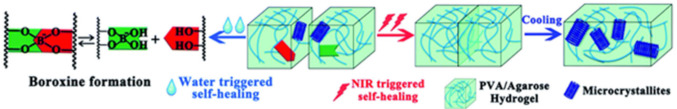
Self-healing mechanisms of the PVA/PDA/MWCNT hydrogel when exposed to NIR and water. Reproduced from [183] with permission from The Royal Society of Chemistry.

**Figure 33 polymers-12-02594-f033:**
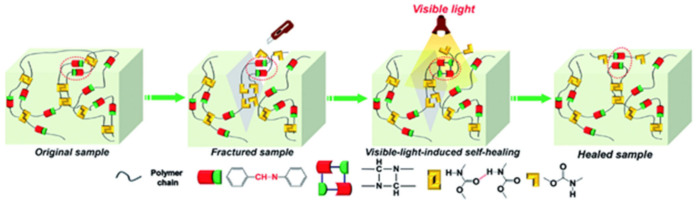
Diagram depicting the healing process of PU polymers. Reproduced from [185] with permission from The Royal Society of Chemistry.

**Figure 34 polymers-12-02594-f034:**
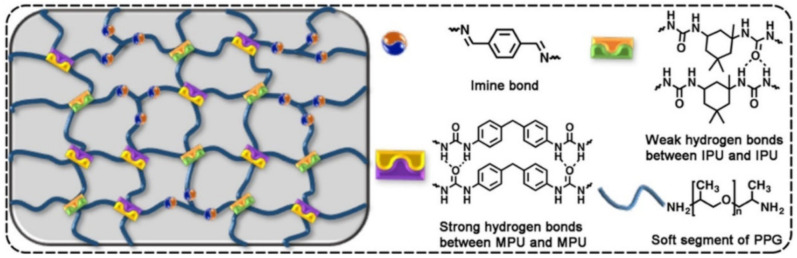
Block diagram of the PU network, which contains both dynamic imine bonds and hierarchical H-bonds. Reprinted with permission from [186]. Copyright (2020) American Chemical Society.

**Figure 35 polymers-12-02594-f035:**
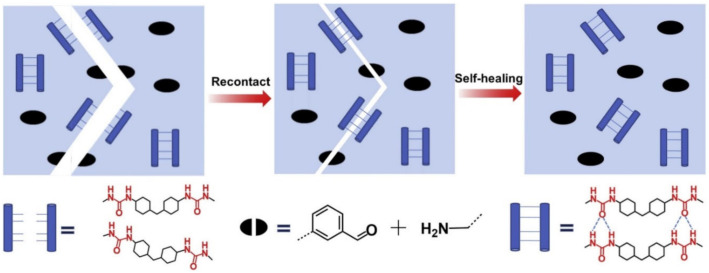
Schematic diagram of the self-healing mechanism. Reproduced from [188] with permission from Elsevier.

**Figure 36 polymers-12-02594-f036:**
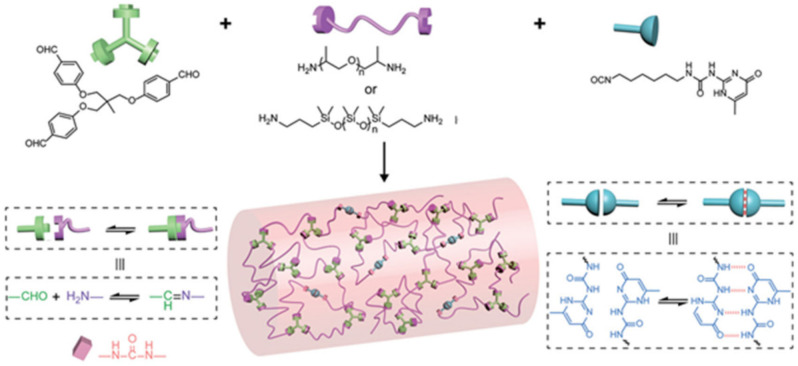
Chemical structure of SHPs cross-linked by dynamic imine bonds and UPy. Reproduced from [189] with permission from John Wiley and Sons.

**Figure 37 polymers-12-02594-f037:**
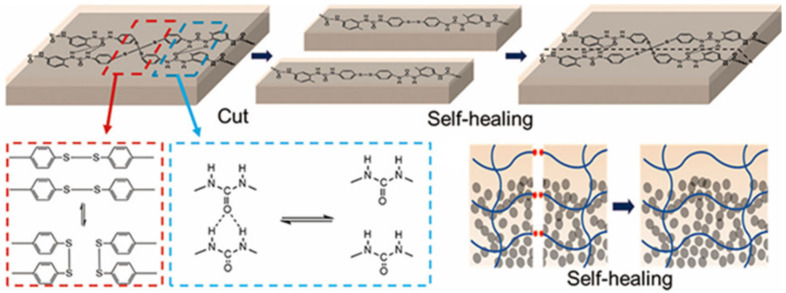
Self-healing process of the developed SHP. Reprinted with permission from [190]. Copyright (2020) American Chemical Society.

**Figure 38 polymers-12-02594-f038:**
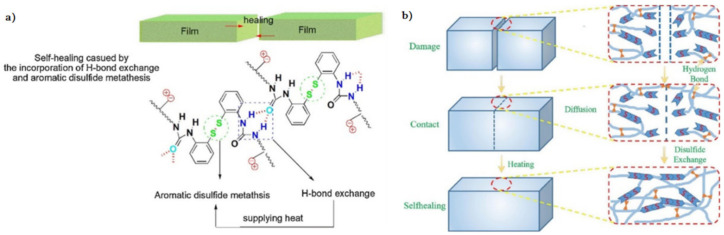
(**a**) A proposed schematic illustration of the self-healing of 2-aminophenyl disulfide-based PU. Reproduced from [191] with permission from Elsevier. (**b**) Schematic representation of the self-healing mechanism of PU materials. Reproduced from [192] with permission from Elsevier.

**Figure 39 polymers-12-02594-f039:**
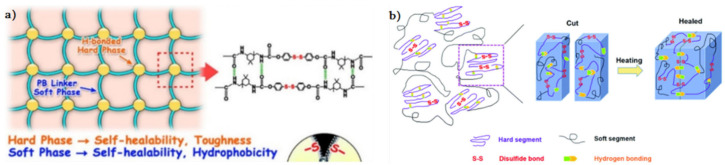
(**a**) Ideal network structure with soft phase and hard phase. Reprinted with permission from [195]. Copyright (2020) American Chemical Society. (**b**) The healing mechanism of SHPs based on disulfide and H-bonds. Reproduced from [196] with permission from The Royal Society of Chemistry.

**Figure 40 polymers-12-02594-f040:**
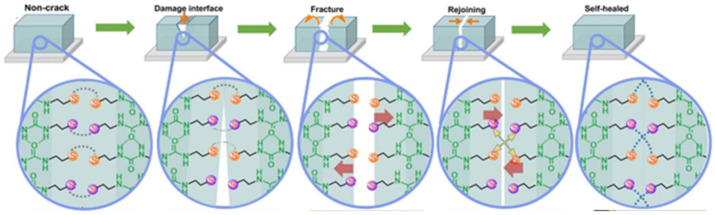
Schematic representation of a self-healing process with a disulfide bond and an H-bond. Reprinted with permission from [197]. Copyright (2020) American Chemical Society.

**Figure 41 polymers-12-02594-f041:**
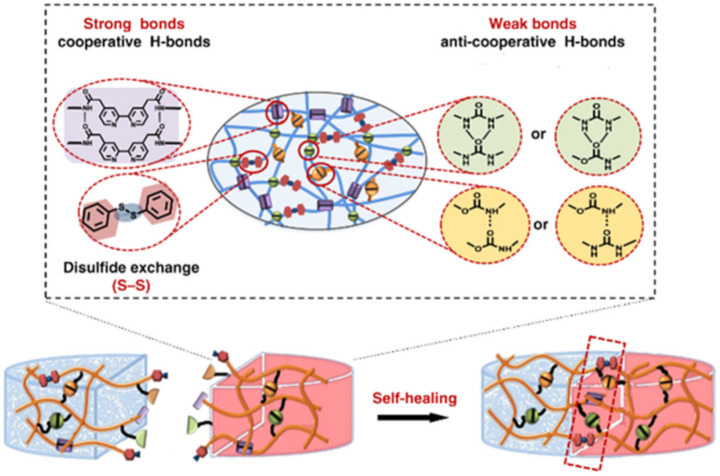
The proposed ideal SHP structure based on strong crosslinking H-bonds, weak crosslinking H-bonds and disulfide metathesis. Reprinted from [199].

**Figure 42 polymers-12-02594-f042:**
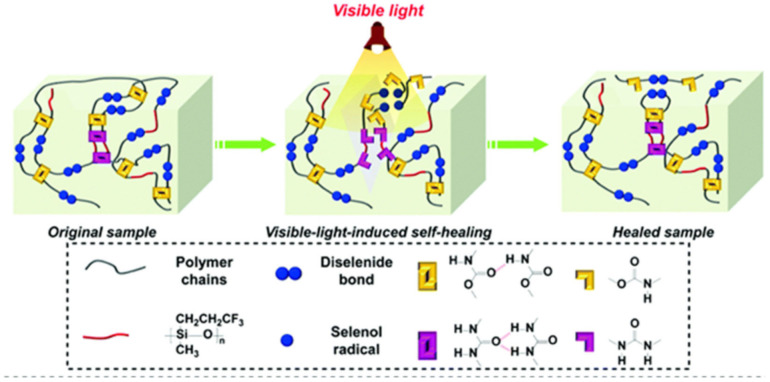
Cartoon representation of the healing process of polymer films in visible light. Reproduced from [201] with permission from The Royal Society of Chemistry.

**Figure 43 polymers-12-02594-f043:**
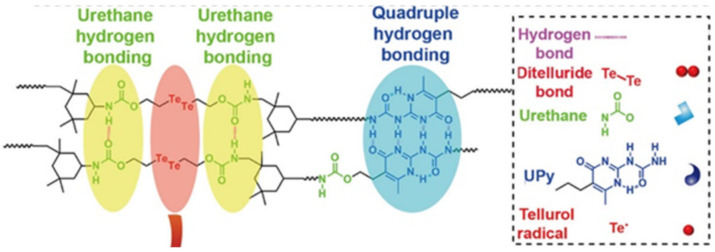
Proposed interactions formed between the macromolecule chains. Reprinted with permission from [203]. Copyright (2020) American Chemical Society.

**Figure 44 polymers-12-02594-f044:**
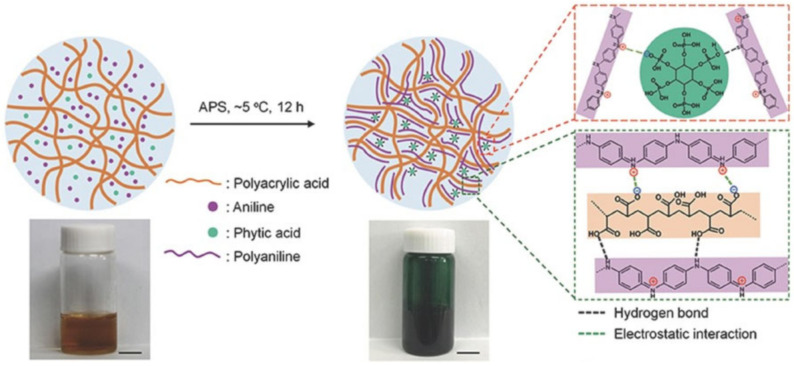
Diagram showing the synthesis of a ternary polymer composite and the corresponding interactions between PANI chains with PAA and phytic acid (scale bar = 1 cm). Reproduced from [205] with permission from John Wiley and Sons.

**Figure 45 polymers-12-02594-f045:**
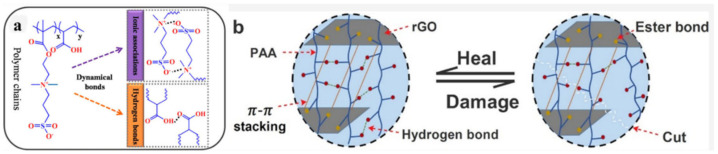
(**a**) Schematic representation of dynamical bonds between cross-linked polymer chains. Reprinted with permission from [207]. Copyright (2020) American Chemical Society. (**b**) Mechanism of self-healing of PAA-rGO elastomers. Reprinted with permission from [208]. Copyright (2020) American Chemical Society.

**Figure 46 polymers-12-02594-f046:**
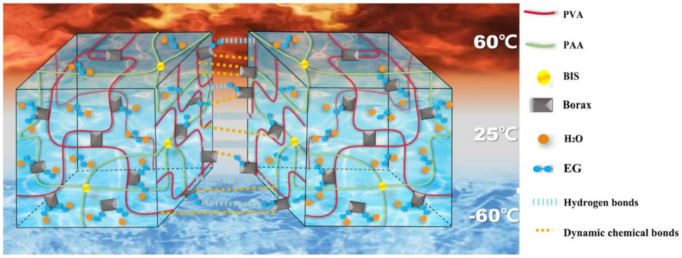
Self-healing mechanisms of the gel. Reprinted with permission from [212]. Copyright (2020) American Chemical Society.

**Figure 47 polymers-12-02594-f047:**
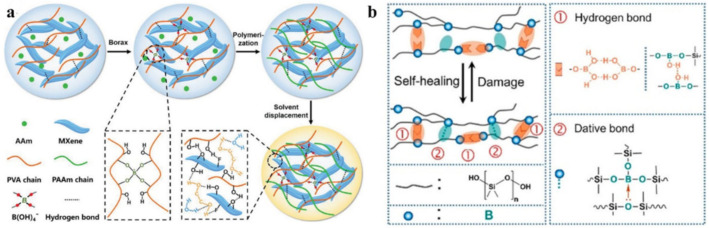
(**a**) Schematic illustration representation of SHP fabrication. Reproduced from [214] with permission from John Wiley and Sons. (**b**) Illustration of the polymer’s self-healing mechanism. Reprinted with permission from [215]. Copyright (2019) American Chemical Society.

**Figure 48 polymers-12-02594-f048:**
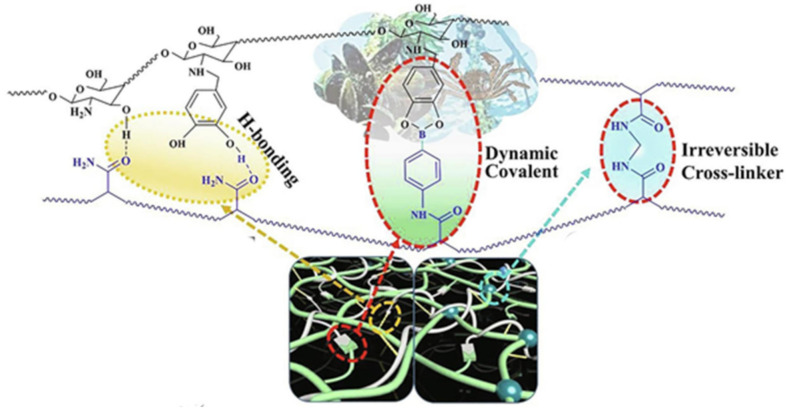
H-bond, dynamic covalent bonds and chemical cross-linkers in the cross-linking networks of chitosan containing catechol groups (black) and PAM (blue). Reproduced from [216] with permission from Elsevier.

**Figure 49 polymers-12-02594-f049:**
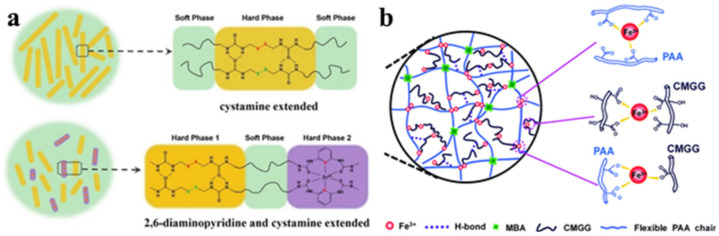
(**a**) Illustration of phase morphology for PU elastomers. Reproduced from [219] with permission from Elsevier. (**b**) Schematic representation of the formation of H-bonds and multiple M–L bonds in hydrogels, where CMGG is carboxymethyl guar gum. Reproduced from [220] with permission from The Royal Society of Chemistry.

**Figure 50 polymers-12-02594-f050:**
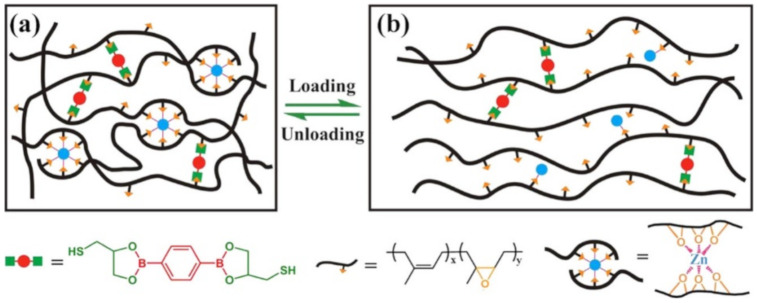
(**a**) Networks with dual cross-links of dynamic boronic ester bonds and non-covalent Zn^2+^–O coordination bonds. (**b**) Reversible breaking and reforming of Zn^2+^–O coordination bonds during loading-unloading tests. Reprinted with permission from [221]. Copyright (2019) American Chemical Society.

**Figure 51 polymers-12-02594-f051:**
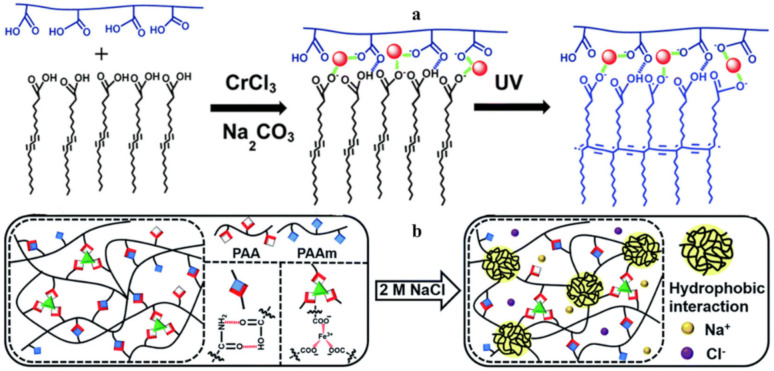
(**a**) Synthesis of the polydiacetylene–PAA–Cr^3+^ elastomer. The PAA network is shown in green, and diacetylene monomers and polydiacetylene are shown in black and blue, respectively. The red spheres correspond to the Cr(H_2_O)_6_^3+^ complex. Reproduced from [222] with permission from The Royal Society of Chemistry. (**b**) Schematic representation of the process of preparing self-healing hydrogels. Reproduced from [223] with permission from The Royal Society of Chemistry.

**Figure 52 polymers-12-02594-f052:**
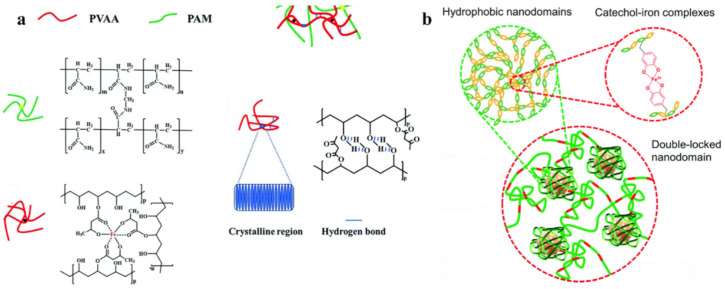
(**a**) Physically cross-linked H-bonds and crystals of aggregated polyvinyl alcohol acetoacetate (PVAA) domains in Fe^3+^/PVAA-PAM hydrogels. Reproduced from [224] with permission from The Royal Society of Chemistry. (**b**) Rigid double-locked hydrophobic nanodomains are composed of multiple iron–catechol complexes and are surrounded by a flexible polymer matrix (PEG). Reproduced from [225] with permission from The Royal Society of Chemistry.

**Figure 53 polymers-12-02594-f053:**
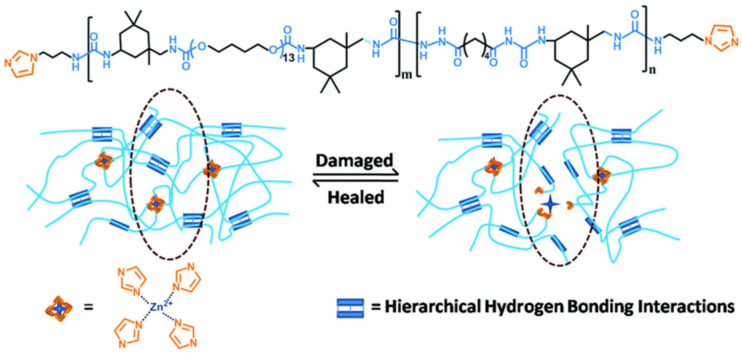
Chemical structure of PU and SHP healing process. Reproduced from [227] with permission from The Royal Society of Chemistry.

**Figure 54 polymers-12-02594-f054:**
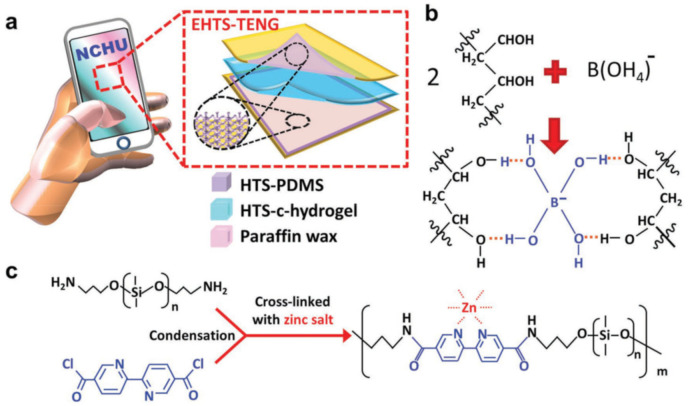
(**a**) Applications and exploded view of TENG. Reactions to form of (**b**) HTS-c-hydrogel and (**c**) HTS-PDMS. EHTS is fully self-healable, transparent and stretchable. Reproduced from [40] with permission from John Wiley and Sons.

**Figure 55 polymers-12-02594-f055:**
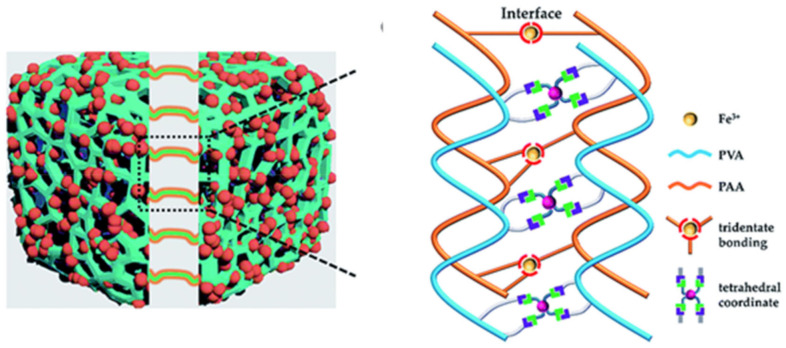
Schematic representation of the self-healing mechanism. Reproduced from [234] with permission from The Royal Society of Chemistry.

**Figure 56 polymers-12-02594-f056:**
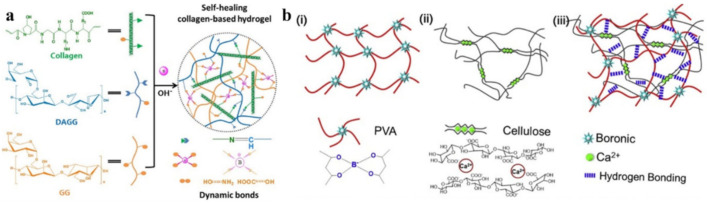
(**a**) Mechanism of collagen-guar gum hydrogel construction, including imine bonds, diol-borate ester bonds, and supramolecular interactions. Reproduced from [235] with permission from John Wiley and Sons. (**b**) Structure of the hydrogel network: (**i**) PVA hydrogel, (**ii**) cellulose nanofibril, and (**iii**) diagram of the PVA/cellulose nanofibril hydrogel network. Reproduced from [236] with permission from Elsevier.

**Figure 57 polymers-12-02594-f057:**
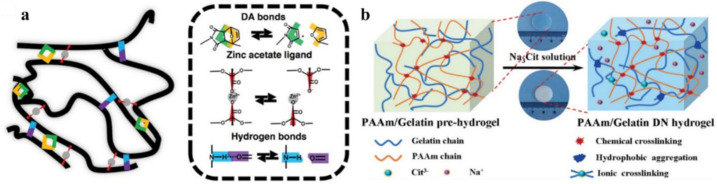
(**a**) Hybrid dynamic network containing triple dynamic bonds, including reversible covalent bonds, zinc–ligand and H-bonds. Reproduced from [238] with permission from John Wiley and Sons. (**b**) Schematic representation of PAAm-Gelatin DN hydrogel. Reproduced from [239] with permission from John Wiley and Sons.

**Figure 58 polymers-12-02594-f058:**
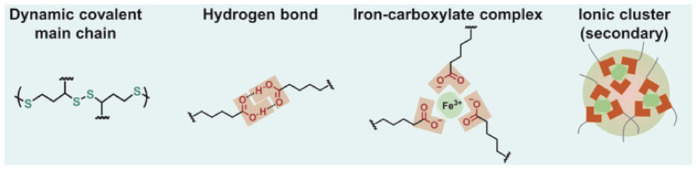
Diagrammatic presentation of the existing four types of dynamic combinations in the network. Reproduced from [241] with permission from John Wiley and Sons.

**Figure 59 polymers-12-02594-f059:**
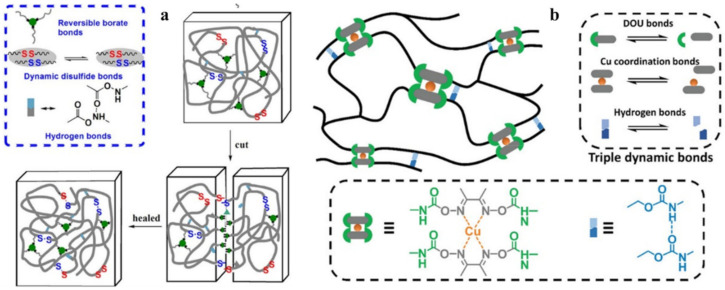
(**a**) Schematic diagram of the PU healing mechanism. Reprinted with permission from [242]. Copyright (2020) American Chemical Society. (**b**) Schematic structure of an elastomer, where DOU is dimethylglyoxime. Triple dynamic bonds, including reversible covalent (green arc and gray stick), M–L (gray stick and orange circle) and H-bonds (blue polygons), built hybrid dynamic networks. Reproduced from [243] with permission from John Wiley and Sons.

**Figure 60 polymers-12-02594-f060:**
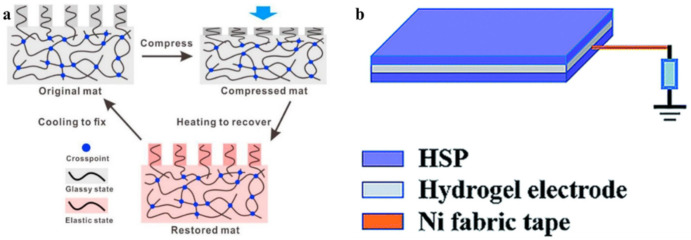
(**a**) Schematic depiction of recovery process of microstructural shape memory PU upon heating. Reprinted with permission from [254]. Copyright (2020) American Chemical Society. (**b**) Structural diagram of TENG based on healing and shape-memory dual-functional polymer (HSP). Reprinted from [255].

**Figure 61 polymers-12-02594-f061:**
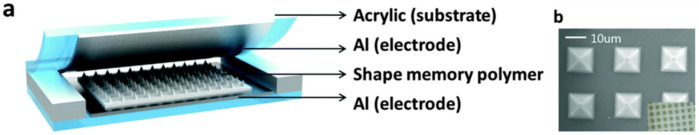
(**a**) Diagram of the structure of TENG. (**b**) Pyramidal pattern of PU on the triboelectric layer of PU. Reproduced from [256] with permission from The Royal Society of Chemistry.

**Figure 62 polymers-12-02594-f062:**
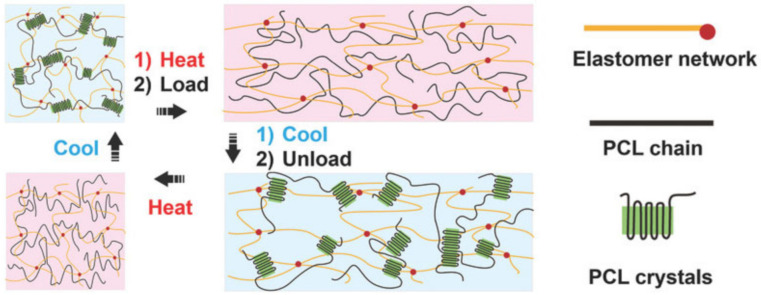
Diagrams showing the mechanism of shape memory behavior. Reproduced from [259] with permission from John Wiley and Sons.

**Figure 63 polymers-12-02594-f063:**
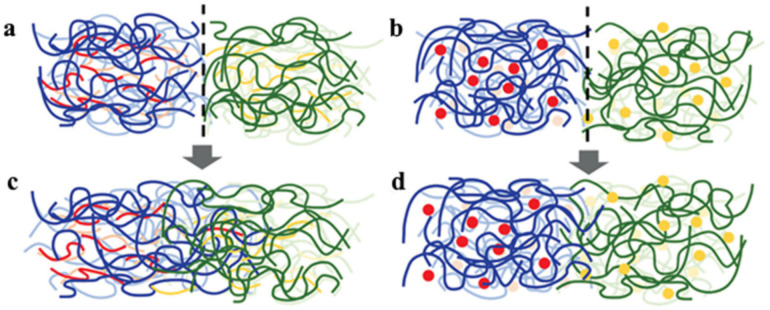
Schematic diagrams of PU/epoxy oligomer (**a**) before and (**b**) after healing and PU/epoxy with bis(4-maleimidophenyl)methane (**c**) before and (**d**) after healing. Reproduced from [268] with permission from John Wiley and Sons.

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
