# Peer review of "Basic Approaches to the Design of Intrinsic Self-Healing Polymers for Triboelectric Nanogenerators"

_polymers, 2020, doi:10.3390/polym12112594_

Round 1

Reviewer 1 Report

The authors of the manuscript “Basic approaches to the design of intrinsic self-healing polymers for triboelectric nanogenerator” review the triboelectric nanogenerator with utilization of self-healing polymers. Given that the use of novel materials to the TENG is considered as a promising approach to develop next generation TENG, the review is timely to be reported for its further development with superior functionality. I recommend this manuscript to be published after addressing the following minor points:

  1. Readers can easily accept the advantageous aspect of utilization of self-healable polymers (SHP) in the TENG but there is no explicit explanation and emphasis on specific beneficial points when SHP is utilized as a part of the TENG. Although the explanation on beneficial characteristics of the SHP are explained well, to enhance the completeness of the manuscript, specific beneficial points of the SHP when it is utilized as a part of the TENG should be properly emphasized from introduction to conclusion several times. For example, the specific environment or condition where the use of SHP becomes crucial can be listed or commented with the advantages.

  2. To enhance the readability, the authors should explain more detail about the positive effect of self-healing materials when it is utilized as an electrode material or direct contact materials in the TENG. For example, in Figure 1, the author should properly explain the reason why the output performance of TENG can be significantly improved compared to that with the traditional copper foil as an electrode.
  3. I understand that the current review deals about the SHP but in some part, I cannot find correlation of the explanation with the TENG. Some parts seem like to be written for explanation about SHP itself not for explanation about SHP with the viewpoint of the TENG. I recommend the authors to add more explanation about the TENG in each section for strengthen the connection between the TENG and the SHP, which is explained in that specific part.
  4. The authors should cite more references regarding TENG to broaden the applicability and to strengthen the impact of the manuscript. In particular, recently, considering that the use of novel mechanical designs to significantly increase the output performance of the TENG, the recommended references with combination of the idea in this review as use of novel materials such as SHP would make the great synergetic effect on its functional enhancement. In this regard, I recommend the authors to additionally cite following papers and the papers should be properly cited in introduction with comments that not only materials scientific aspect, but also mechanics and manufacturing processes are both crucial to be investigated for further development of the TENG. “high quality electret based triboelectric nanogenerator for boosted and reliable electrical output performance, IJPEM-GT, 2020”, “Cold-rolled robust metal assisted triboelectric nanogenerator for extremely durable operation, Extreme Mechanics Letters, 2020”, “Universal biomechanical energy harvesting from joint movements using a direction-switchable triboelectric nanogenerator, Nano Energy, 2020”, “Triboelectric signal generation and its versatile utilization during gear-based ordinary power transmission, Nano Energy, 2020”, “Monocharged electret based liquid-solid interacting triboelectric nanogenerator for its boosted electrical output performance, Nano Energy, 2020”, “Development of a high-performance handheld triboelectric nanogenerator with a lightweight power transmission unit, Advanced Materials technologies, 2020”.

Reviewer 2 Report

Triboelectric nanogenerators (TENGs) as a revolutionary system for harvesting mechanical energy have demonstrated high vitality and great advantage, which open up great prospects for application in various areas of the society of the future. This review presents and evaluates self-healing polymers (SHPs) for TENGs. Such as the non-covalent (hydrogen bond, metal–ligand bond), covalent (imine bond, disulfide bond, borate bond) and multiple bond-based SHPs in TENGs has been performed. There are some issues should be addressed.

  1. The authors have mentioned the approaches for producing TENG with high output power. However, is it suitable to say the choice of tribomaterial is the most effective approach? A lot of approaches, such as the charge pump and circuits, can also obtain the high output power for TENG.
  2. This review mainly focused on the different SHPs and the healing efficiency for TENGs. However, the healing cycles and the stability are also important for the self-healing TENGs.
  3. Also, whether different healing approaches have their own advantages and effects for producing TENG with high output power?
  4. Most SHPs require external stimuli (heat, moisture, light, etc.) to trigger the healing process. The SHPs with spontaneous healing process must be improved for the future.
  5. Are there any detail potential applications for these self-healing TENGs?
